# Preparation, Characterization and Evaluation of the Anti-Inflammatory Activity of Epichlorohydrin-β-Cyclodextrin/Curcumin Binary Systems Embedded in a Pluronic^®^/Hyaluronate Hydrogel

**DOI:** 10.3390/ijms222413566

**Published:** 2021-12-17

**Authors:** Ana-María Fernández-Romero, Francesca Maestrelli, Sara García-Gil, Elena Talero, Paola Mura, Antonio M. Rabasco, María Luisa González-Rodríguez

**Affiliations:** 1Department of Pharmacy and Pharmaceutical Technology, Faculty of Pharmacy, Universidad de Sevilla, C/Prof. García González 2, 41012 Seville, Spain; anaferrom2@alum.us.es (A.-M.F.-R.); amra@us.es (A.M.R.); 2Department of Chemistry “Ugo Schiff” (DICUS), University of Florence, Via Schiff 6, Sesto Fiorentino, 50019 Florence, Italy; francesca.maestrelli@unifi.it (F.M.); paola.mura@unifi.it (P.M.); 3Department of Pharmacology, Faculty of Pharmacy, Universidad de Sevilla, C/Prof. García González 2, 41012 Seville, Spain; saragarciagil2307@gmail.com (S.G.-G.); etalero@us.es (E.T.)

**Keywords:** curcumin, cyclodextrin, epichlorohydrin-β-cyclodextrin, pluronic, hydrogel, inflammatory disease, psoriasis

## Abstract

Curcumin (Cur) is an anti-inflammatory polyphenol that can be complexed with polymeric cyclodextrin (CD) to improve solubility and bioavailability. The aim of the present work was to prepare a CurCD hydrogel to treat inflammatory skin conditions. Epichlorohydrin-β-CD (EpiβCD) was used as polymeric CD. To characterize the binary system, solid-state and in-solution studies were performed. Afterwards, an experimental design was performed to optimize the hydrogel system. Finally, the CurEpiβCD hydrogel system was tested for anti-inflammatory activity using a HaCat psoriasis cell model. Co-grinded Cur/EpiβCD binary system showed a strong interaction and Curcumin solubility was much improved. Its combination with Pluronic^®^ F-127/hyaluronate hydrogel demonstrated an improvement in release rate and Curcumin permeation. After testing its anti-inflammatory activity, the system showed a significant reduction in IL-6 levels. Hydrogel-containing CurEpiβCD complex is a great alternative to treat topical inflammatory diseases.

## 1. Introduction

Curcumin (Cur) is the main active ingredient of turmeric (*Curcuma longa*). This spice has been used in some Asian countries as traditional medicine for multiple purposes, such as worm treatment, swelling, improving digestion, etc. It is generally considered safe and efficient as treatment [1].

Curcumin has many therapeutic applications due to its anti-inflammatory and antioxidant properties. Its anti-inflammatory activity derives from Curcumin capability to modulate TNF-α, IL-1β and IL-6 expressions and inhibit COX-2, NF-κB and MAPK pathways [2]. Its antioxidant activity derives from its structure. Curcumin is an heptadienone with highly activated central CH_2_ prone to reaction. In addition, the phenolic OH is also susceptible to reaction, losing a proton. In both cases, when an oxidant agent reacts with Curcumin, the molecule loses a proton, creating a radical that is stabilized by delocalization [2,3].

Nonetheless, Curcumin has an extremely low solubility and bioavailability that hinders its therapeutic use. Many approaches to overcome this issue have been reported, such as micelles [4], nanoemulsions [5], nanoparticles [6] and cyclodextrins [7].

Cyclodextrins (CDs) are cyclic oligosaccharides consisting of α-1,4 linked glucopyranose units listed as GRAS (Generally Recognized as Safe) excipients by the Food and Drug Administration (FDA) and usually added to food and pharmaceutical products [8]. CDs are useful for improving low aqueous solubility of lipophilic drugs since they have the ability to form inclusion complexes with hydrophobic drugs in aqueous solutions, hosting them into their hydrophobic cavity. The main driving forces of the complex formation are van der Waals forces and hydrophobic interactions without any covalent bonding, and the solubilizing effect is given by the hydrophilic outer surface of the CD molecules. Natural CDs contain 6, 7 and 8 glucose units and are named α, β and γ cyclodextrin, respectively. β-cyclodextrin (βCD) is able to form complexes 1:2 mol/mol with Curcumin, thus, improving its solubility and its pharmacological effect [9,10,11]. On the other hand, since the low solubility and low complexing ability of natural CDs, several derivatives have been synthesized over the years. Tønnesen et al., (2002) in a comparative study with various, CDs showed that the complexes formed were of type 1:1 mol/mol and that randomly methylated βCD and hydroxypropyl-γ-CD had the highest complexing ability toward Curcumin [12]. Among the derivatives, the soluble epichlorohydrin polymer derivative of β-CD (EpiβCD) showed a great ability on improving drug solubility [13,14] and oral bioavailability [15,16] after its incorporation in drug formulation. A recent study showed the enhanced antioxidant and antiproliferative activity of curcumin obtained by complexation with a purposely synthesized EpiβCD [17]. On the other hand, the preparation method of binary systems Cur/CD also has a certain importance in determining the performance of the final product [18]. With these premises, EpiβCD has been proposed as a great candidate to form complex with Curcumin in treatment of multiple diseases.

The aim of the present work was to design a topical formulation to serve as treatment for skin inflammatory diseases, such as psoriasis. Several recent studies have been reported about the combination of cyclodextrins complexes and hydrogel [19,20,21] also for curcumin administration [22]. In this case, a gel formulation based on the combined use of Pluronic^®^ F127 and EpiβCD was developed and used to exploit benefits connected with their inherent properties, namely, beneficial Curcumin stabilization in this polymer and the complexing power and transdermal penetration enhancer of EpiβCD [23], in order to obtain an optimal local delivery of Curcumin. Finally, to demonstrate the anti-inflammatory activity of the proposed system, a HaCat cell psoriatic model was employed, which was selected due to previous experience in our research laboratory [24].

## 2. Results

### 2.1. Phase Solubility Studies

In order to evaluate the complexing and solubilizing abilities of EpiβCD towards Curcumin, phase solubility studies were performed in phosphate buffer pH 5. This pH was selected as a compromise between Curcumin stability and skin pH [25]. Curcumin solubility increased linearly with increasing EpiβCD concentrations (Figure 1a), indicating the formation of a soluble inclusion complex. The stability constant, calculated by taking the CD repeating unit as its molecular weight, was 4.5 times higher than with the parent CD. This value (6.186 M^−1^) is higher than those obtained by Tønnesen et al., (2002) with other derivatives [12]. The superior complexing and solubilizing efficiency of polymeric CD derivatives can be attributed to their higher aqueous solubility, but also to their polymeric structure, which allowed an efficient cooperation of adjacent CD cavities for interaction with the drug [26]. Since no deviations from linearity were observed in phase solubility diagrams, the formation of higher order complexes in the CD concentration range studied can be excluded. Taking into account these results, binary systems were prepared at 1:10 *w*/*w* Cur:EpiβCD ratio.

### 2.2. Solid-State Studies

In order to evaluate the effect of the preparation techniques, different binary systems of Curcumin with EpiβCD were prepared and submitted to the DSC analysis. As shown in Figure 1b, Curcumin presents a melting peak at 179.72 °C with a melting enthalpy of 95.42 J/g, which is in accordance with bibliography [27], while EpiβCD showed an amorphous profile with a broad endothermic band from 40 to 120 °C associated with water loss. Curcumin melting peak was almost unchanged in the physical mixture (PM) with EpiβCD, indicating the absence of solid-state interactions between the components. On the contrary, the drug melting peak intensity gradually decreased as a function of the technique used, with the trend PM = kneading (KN) < colyophilizated (COL) < coevaporated (COE), and it completely disappeared in the co-grinding (GR). This indicated that the co-grinding procedure was the most successful to induce effective solid-state interactions between the components, resulting in complete Curcumin amorphization and/or complexation. Solid-state interactions between the components were further investigated by XRD and FTIR analysis. The XRD spectra of pure drug and EpiβCD, the PM and different binary systems samples are presented in Figure 1c. Pure Curcumin showed sharp diffraction peaks at 2ϴ between 9 and 29 [27] indicating its crystalline nature, while the pattern of EpiβCD was typical of a completely amorphous substance. Diffraction patterns of the PM, KN and COL products almost corresponded to the superimposition of those of the plain components, where the peaks at 9, 15 and 18° were still present, confirming a limited interaction between the components. COE showed, instead, a marked reduction of the drug diffraction peaks intensity and GR sample presented a completely amorphous profile. FT-IR analysis further confirmed the above results (Figure 1d). In fact, the peaks of Curcumin at 1627–1602 cm^−1^ attributed (more detail, see Appendix A) to the stretching of the benzene ring of Curcumin [28], covered all the binary systems by the broad band of the CD, and appeared shifted at 1583 cm^−1^ in the GR, thus, confirming the deeper interaction obtained with this preparation technique.

### 2.3. Dissolution Rate Studies

The higher effectiveness of GR in inducing drug–CD interactions was further corroborated by results of dissolution studies. The dissolution profiles are reported in Figure 2, where it is visible that all binary systems with EpiβCD exhibited improved dissolution properties with respect to drug alone (Curcumin), but a clear influence of the preparation method on the performance of the final product can be noticed. In particular, it is evident that GR system exhibited the highest dissolution rate (85 mg/L of dissolved Curcumin after 5 min) and the highest solubility (100 mg/L) followed by COE product (20 mg/L of dissolved Curcumin after 10 min) and then by KN and COL products. The overall results indicated that the GR was the most efficacious preparation technique to obtain complete drug amorphization and/or complex formation, and maximum improvement of drug dissolution rate. Moreover, GR is a fast, highly efficient, convenient, versatile, sustainable and eco-friendly solvent-free method for obtainment of drug–cyclodextrin binary systems [29], and thus the co-ground system was chosen as the most effective product to introduce in the final formulation.

### 2.4. Optimization of Gel Formulation

The aim of the present work was to design a topical formulation to serve as treatment for skin inflammatory diseases, such as psoriasis. In this case, a hydrogel was selected as hydrophilic vehicle. To formulate it, four polymers were chosen: Pluronic^®^ F-127, Carbopol^®^ 940, chitosan and sodium hyaluronate. Pluronic^®^ F127 is a poloxamer able form rigid gels at different concentrations and temperatures, being the temperature dependent of the concentration of the polymer. In order for it to reticulate at 32 °C (skin temperature), Pluronic^®^ F-127 concentration should be at least of 15% *w*/*v* [30]. Contrary, this concentration is too high for Carbopol^®^ 940, chitosan and sodium hyaluronate. For this reason, Pluronic^®^ F-127 was kept constant throughout the experimental design.

An orthogonal Taguchi L9 array was performed and the effects of Curcumin concentration (added as GR with EpiβCD), type of polymer, polymer concentration and ratio of Pluronic^®^/polymer over % of drug released at 6 h, % of permeated drug at 6 h, pH and apparent viscosity of the gel, were analyzed.

Results of each experiment and their experimental conditions are shown in Table 1. In this case, permeation varied from 0.0491 to 0.3394%, drug release from 0.6183 to 3.2814%, gel viscosity from 0.0972 to 3.8998 Pa∙s, and pH from 4 to 5.

After analyzing all results, statistical significance of these factors on the evaluated responses was clarified with ANOM (Analysis of Means) of the main factors on each response (data not shown). Moreover, results were analyzed by ANOVA (Analysis of Variance) (Table 2) and contributions of each response were described in a Pareto chart (Figure 3). This type of chart facilitates information about how much each parameter contributes to the response and the sign of each contribution. In this case, all responses showed significant results.

Apparent viscosity of the system was affected by all factors studied (Figure 3a). The polymer concentration had a positive effect on viscosity, indicating that at higher concentration, viscosity will also be higher. The type of polymer had an interesting effect on this parameter. The effect was negative when chitosan was compared with hyaluronate but it was positive when compared to Carbopol^®^, which indicated that Carbopol^®^ increased the viscosity more than the other two polymers. As expected, the ratio also affected, in all cases in a positive way, indicating that the more polymer was added, the higher the viscosity. Curcumin concentration also had a curious behavior, as it was negative when compared with the medium level (0.3 mM of Curcumin) but positive when extreme levels were compared.

Regarding pH, only two factors contributed significantly to its behavior: the proportion of Pluronic^®^/polymer and the type of polymer used to create the hydrogel (Figure 3b). Interestingly, only the ratios 20:80 and 50:50 affected the pH, increasing it with polymer diminution. Probably, at 80:20 ratio, the amount of polymer compared to the amount of Pluronic^®^ is not enough to change the pH. Type of polymer also affects the pH, which is consistent with previous knowledge. Pluronic^®^ solution has a pH of 6–7, while sodium hyaluronate and chitosan had a pH of 7 and below 6, respectively. Logically, the addition of chitosan will lower the pH while sodium hyaluronate will increase it. Carbopol^®^, on the other hand, needs to be neutralized to reticulate, leading to a pH of 7 that will not affect significantly Pluronic^®^ pH.

In the case of in vitro permeation (Figure 3c), three actors affected the result: the polymer concentration, the Curcumin concentration and the ratio of Pluronic^®^/polymer. Curcumin concentration affected negatively the amount of Curcumin able to permeate skin, when low and high levels were compared; however, when both of them were compared with the middle level (0.3 mM), the effect was neutral, so, the permeation was favorable until this drug concentration. As Figure 3c shows, the proportion of Pluronic^®^/polymer affects in a positive way, indicating that the higher the Pluronic^®^ content, more Curcumin can permeate through the membrane. The same behavior was obtained with the polymer concentration (the amount of Curcumin permeated increased as the polymer concentration increased).

The in vitro release behavior of Curcumin was similar to that of permeation except for the contribution of the type of polymer. In this case, the use of sodium hyaluronate provided more Curcumin released than the addition of chitosan. Carbopol^®^ did not affect the release (Figure 3d). After analyzing ANOM and ANOVA data, optimized formulation was proposed. The main objective of this array was to maximize permeation, release and viscosity, while keeping pH close to 5.5. However, as can be seen in the Pareto charts (Figure 3), some factors affecting Curcumin release and permeation are in opposition to those affecting viscosity. This indicates that in order to maximize viscosity, permeation and release will be hindered. For that reason, permeation and release were chosen as more important and favorable responses than viscosity. As a result, optimized gel components were as follows: Curcumin concentration was kept at 0.2 mM, sodium hyaluronate 3% *w*/*v* was chosen as polymer and concentration and the ratio of Pluronic^®^/polymer was set at 80:20.

### 2.5. Characterization Studies

Once the optimized composition of the gel was obtained, physicochemical characterization was carried out as was mentioned in Section 4.9.

#### 2.5.1. Apparent Viscosity, Density and pH

Apparent viscosity was measured only to the hydrogel formulation containing CurEpiβCD GR. The result was 0.135 Pa∙s at 62.65 s. Additionally, representation of shear rate and viscosity was also considered. On it, viscosity diminished with shear rate, indicating that this hydrogel is a non-Newtonian fluid.

Regarding pH and density, they were 6 and 0.979 g/mL, respectively.

#### 2.5.2. Gelation Temperature and Storage and Loss Moduli

Pluronic^®^ F-127 is a poloxamer appreciated for its ability to jellify with temperature. However, the addition of other components to its solution can modify the gelling temperature. For this reason, different solutions of Pluronic^®^ were tested: Pluronic^®^ 17% *w*/*v* water solution (PluW), Pluronic 17% *w*/*v* water/ethanol solution (PluWE), PluWE containing EpiβCD (PluWE-Epi), optimized gel containing only EpiβCD (Epi-Gel) and optimized gel loaded with CurEpiβCD GR binary system (CurEpiβCD-Gel). Additionally, sodium hyaluronate (Hyal) was also analyzed for comparation purposes.

To study changes in the gelation temperature, two tests were taken: inversion tube and the intersection between storage (G′) and loss (G″) moduli. To obtain the gelling temperature via G′ and G″ crossover, both moduli should be equal. If G′ = G″, loss tangent (tanδ) equals 1. Consequently, the range in which this tangent is 1 or near 1 was selected as the gelling temperature.

Table 3 shows the results of both assays. As can be seen, both tests resulted in different gelatin temperatures. For PluW, inversion test resulted in gelation above 35 °C and returned to solution state after 44 °C. The range obtained by tan (δ) calculation indicated a narrower range (38.05–41.06 °C). In both cases, the range is in accordance with bibliography [31]. Comparing PluWE and PluW, gelling temperature was lower (15.37 °C) and the system returned to a liquid state at a lower temperature (23.09 °C). The addition of EpiβCD to PluWE resulted in an increase in the gelling temperature in comparison with PluWE, a narrower range was obtained. Optimized gel with and without Curcumin (CurEpiβCD-gel and Epi-Gel, respectively) showed a similar temperature range to that of PluWE, indicating that the addition of sodium hyaluronate can increase gelling temperature. As can be seen in Appendix A, after 5 min at 20 °C, CurEpiβCD-gel and Epi-Gel did not test positive for inversion test, indicating that even after a change in the structure occurs, the optimized hydrogel is not able to form structures as rigid as PluW.

Sodium hyaluronate was also analyzed as a control sample. This component does not have a gelling temperature; however, changes in viscosity with temperature are well known. For this reason, it was also analyzed. Interestingly, an increase in tan (δ) until 1 was detected, followed by a plateau (range mentioned in Table 3), and, finally, a decrease in the parameter.

Regarding loss and storage moduli, the same samples were analyzed. Results are showed in Figure 4. PluW exhibited similar G′ and G″ that increased with temperature until 40 °C, when G″ rapidly decreased while G′ decreased slowly. Both moduli increased again until 65 °C but G′ was higher than G″. The addition of ethanol resulted in a slower increase of G′ and G″, with an initial plateau at 15–20 °C were G″ was higher that G′. At gelation temperature, G′ surpassed G″. A similar behavior was noted in PluWE-Epi sample, Epi-Gel and CurEpiβCD-Gel (Figure 4c–e) performed similarly to PluWE.

Sodium hyaluronate presented a different behavior than Pluronic^®^. In this case, G′ decreased while G″ was almost constant and reach a plateau between 50 and 58 °C, after which G′ abruptly increased. G″ also increased, but not as abruptly as G′.

#### 2.5.3. Curcumin Content

Once the optimized hydrogel was prepared, the amount of Curcumin extracted was quantified. The result showed that a 95.54% of Curcumin can be extracted from the hydrogel.

#### 2.5.4. In Vitro Release and Curcumin Permeation

The optimized hydrogel was then assessed in terms of in vitro Curcumin release and permeation. As comparison, a solution of GR CurEpiβCD was prepared following the same procedure as described in Section 4.7. As previous results showed (Table 2), the percentage of Curcumin released and permeated after 6 h from the hydrogels in the optimization step, indicating slow processes. For that reason, in this section, both studies were performed over the course of 72 h.

As can be seen in Figure 5a, Curcumin release from the solution was completely opposed to that from the gel. At 8 h, the amount of Curcumin released from the CurEpiβCD solution was about 30%, while the hydrogel released only a very low drug amount. Interestingly, later in the experiment, the hydrogel released its content slowly but regularly, while the solution exhibited a degradation process after 48 h.

During the in vitro permeation studies, Curcumin behavior was completely different (Figure 5b). In this case, Curcumin permeated from the CurEpiBCD solution was only about 20% after 48 h and remained almost unchanged at 72 h. In the case of the hydrogel, the amount of Curcumin permeated was similar to that of the solution until 8 h of experiment. Afterwards, the optimized system allowed a strong increase of the amount of permeated Curcumin until 48 h (almost 60%); however, after this point, degradation overcame permeation and the amount of Curcumin permeated at 72 h, decreased (40%).

### 2.6. Stability Studies

Once CurEpiβCD-Gel was characterized, its physical and chemical stabilities were monitored. This assay was performed over the course of three months, following the schedule proposed in Section 4.9.

During this study, the stability of the hydrogel formulation and the Curcumin were analyzed using different parameters. For gel stability, pH was selected as it can exert major influence on Curcumin stability. For Curcumin stability, two parameters were studied: Curcumin content and antioxidant capability, as they seem to be related to its pharmacological effects.

Regarding pH, unloaded gel (Em-Gel) and optimized hydrogel (CurEpiβCD-Gel) kept their pH at 6 up to day 15, when it dropped to 5. Complex solution (CurEpiβCD-Sol) always kept pH at 5.

Curcumin oxidation was measured by means of ABTS assay. This procedure compares the antioxidant activity of any product to that of Trolox. In this case, data should be interpreted as the smaller the EC_50_, the higher antioxidant capability the molecule has. In all cases, Trolox EC_50_ equals one. As Figure 6a shows, in the hydrogel, Curcumin EC_50_ was maintained for 8 days and reduced at day 15 (*p* = 0.0013 EC_50_ day 8 vs. day 15). Then, it stabilized again until day 21, increasing again at day 56 (*p* = 0.0137). This indicates that Curcumin antioxidant capability increased at day 15 and decreased again at day 56. In the case of the solution, EC_50_ was almost constant throughout the experiment as no statistical significance was found between the samples. Comparing the solution and the hydrogel, it was revealed that the hydrogel had less antioxidant capability that solution for the first 15 days for later increase at 21 days (*p* = 0.0239 at day 21 hydrogel vs. solution). At the end of the experiment, the hydrogel exhibited a lower antioxidant capability than the solution (*p* = 0.0225).

Curcumin content in the system had surprising results (Figure 6b). From day 1, Curcumin content in the hydrogel formula was reduced to 60%. Curcumin content kept decreasing during the entire experiment, reaching 20% at day 84. Contrary to expectations, Curcumin solution was more stable over time, decreasing to 65% at 84 days. As both samples contained CurEpiβCD complex, this lack of stability can only be attributed to the hydrogel.

As can be extracted from this information, although Curcumin content was decreasing in the optimized hydrogel, antioxidant activity was not decreasing accordingly. In fact, it was maintained. This can be attributed to Curcumin degradation products. More about this topic will be addressed in the discussion section.

### 2.7. Cell Studies

#### 2.7.1. Cell Viability Studies

The effect of the hydrogel on HaCaT cell viability was measured by using the resazurin proliferation assay. Results from cytotoxicity study showed that none of the tested concentrations affected cell viability (Figure 7). The inhibitory concentration 50 (IC50) (half maximal inhibitory concentration) was above 100 µM at 48 h after treatment.

#### 2.7.2. Effect of CurEpiβCD on IL-6 Production in TNF-α-Stimulated HaCaT Human Keratinocytes

Non-cytotoxic concentrations of CurEpiβCD-Gel were used to determine its effect on the pro-inflammatory cytokine IL-6 in human keratinocytes. As shown in Figure 7, TNF-α stimulation induced a significant increase in the production of IL-6 in HaCaT cells compared to unstimulated control cells (*p* < 0.001). The pre-treatment of cells with CurEpiβCD-Gel at 10 µM showed a significant reduction of IL-6 levels in TNF-α-stimulated HaCaT keratinocytes (*p* < 0.05) (Figure 8).

## 3. Discussion

The aim of the present work was to optimize the formulation of a hydrogel with high content of Curcumin to treat inflammatory diseases. As Curcumin is a polyphenol with extremely low solubility (3 × 10^−3^ µg/mL, [32]), a solubilizing strategy was necessary. In this case, cyclodextrins were employed.

The authors have already used this strategy with 2-hydroxypropyl-βCD [33]. However, although Curcumin solubility increased greatly compared with the raw polyphenol, it was not as high as desired. In addition, EpiβCD has already been tested for topical use, resulting in an enhanced pharmacological efficacy [13]. For these reasons, epichlorohydrin-β-cyclodextrin (EpiβCD) was chosen.

EpiβCD is a polymeric cyclodextrin synthesized by condensation and polymerizing βCD with epichlorohydrin in alkaline medium, creating a non-ionic CD [23]. As mentioned before, its structure allows for a more efficient interaction between drug and polymer. In a previous work, we formulated CurHPβCD 1:2 complex with a Ks of 4.86 M^−1^ [33], which is 1.273 times smaller than Ks obtained for CurEpiβCD complex. This indicates that a stronger bond Cur–CD has formed in the polymeric CD. Moreover, in case of EpiβCD, since the ratio is 1:1 M, less CD is necessary to solubilize Curcumin. On the other hand, also the binary systems Cur/CD preparation method has a certain importance in improving drug solubility [18] since it can affect particle size, powder wettability and drug–CD interaction. Considering phase solubility results, Cur/EpiβCD binary systems have been prepared at 1:10 *w*/*w* ratio with four different methods: co-evaporation (COE), co-grinding (GR), kneading (KN), co-lyophilization (COL) and compared with the simple physical mixture (PM) at the solid state and in solution. Thermal analysis performed with DSC evidenced a disappearance of Curcumin melting peak in the COE and GR. This phenomenon can be attributed to a drug amorphization and/or complexation but since it can be due to the thermal treatment it must be confirmed by other technique. XRD confirmed the DSC results showing that, while the PM, KN and COL products showed the presence of the Curcumin diffraction peaks, in the GR sample, an amorphous profile is evident. On the other hand, COE showed a reduction of peak intensity but not a complete disappearance, demonstrating an incomplete amorphization and/or complexation of the Curcumin in the CD with this technique. FT-IR analysis further confirmed the above results, showing a deeper interaction of the CD with the benzene ring of Curcumin [28] in case of GR treatment of the sample. Finally, the dissolution rate studies showed a relationship between the drug amorphization/complexation with the CD obtained with the different preparation techniques and drug solubility and dissolution rate. In fact, the GR system exhibited the highest solubility and dissolution rate followed by COE product, then by KN and COL products. The overall results indicated that the GR was the most efficacious to obtain complete drug amorphization and/or complex formation, and maximum improvement of drug dissolution rate. Given its simplicity, convenience, speed, versatility, sustainability and effectiveness [29], the grinding technique was, thus, chosen to obtain the Cur/EpiβCD binary system to introduce in the final formulation.

As mentioned previously, Curcumin has anti-inflammatory properties. The use of Curcumin topically is well known [34,35], as it has already been employed for psoriasis [36,37], edema [38], etc. In order to topically apply the Cur/EpiβCD binary system, a suitable formulation was necessary. Due to psoriasis has been chosen as inflammatory model, a hydrogel was selected as it can be also favorable for other applications and diseases.

Four different gelling agents were selected for this purpose: Pluronic^®^ F-127, Carbopol^®^ 940, high molecular weight chitosan and sodium hyaluronate.

Pluronic^®^ is a poloxamer known for its ability to reticulate at 37 °C. This property can be interesting for topical use as it can create occlusion and enhance drug permeation. In addition, Mondal et al., (2016) demonstrated that some forms of Pluronic^®^ can reduce Curcumin degradation due to its inclusion into the hydrophobic core of the polymer [39]. Chitosan is a polysaccharide composed of *N*-acetyl-D-glucosamine and D-glucosamine units linked by 1-4-β-glycosidic bonds. Its structure provides −NH_2_ groups that are protonated at low pH [40], favoring interaction with the negatively charged stratum corneum [41]. In addition, chitosan has already proved its compatibility with EpiβCD [42]. In this case, high molecular weight chitosan was employed because of its higher viscosity. Carbopol^®^ is a poly(acrylic acid) that is able to interact with βCD and HPβCD by forming hydrogen bonds [43]. Sodium hyaluronate is an anionic polysaccharide known for its mucoadhesive and permeation enhancing properties [44].

Once components were selected, the optimization was performed applying a Taguchi L9 array. Responses were analyzed with ANOM and ANOVA, and the optimum composition was selected. In this array, the objective was to maximize viscosity, % Curcumin released and % Curcumin permeated at 6 h while maintaining pH in the 4–5 range. However, some factors affecting viscosity were in complete opposition with release and permeation. As a compromise solution, those parameters that will maximize release and permeation and, at the same time, keep viscosity at an intermediate level were selected. Viscosity was maximized with Carbopol^®^. However, the use of this polymer would hinder Curcumin release. This is contradictory with previous works carried out with this polymer [45,46,47]. This carbomer did not affect permeation of Curcumin, which is also contrary to bibliography. Since, in these papers, Curcumin is not complexed with a polymeric CD like EpiβCD, its presence can be the reason behind these surprising results [48]. Nonetheless, the interaction between Curcumin and Carbopol^®^ and its implication on Curcumin release are not fully studied. Regarding the ratio of Pluronic^®^/polymer, a ratio of 80:20 enhanced release and permeation but minimized viscosity. This occurs because any of the polymers are more viscous than Pluronic^®^; hence, lower quantities of this poloxamer in the formula would logically reduce viscosity.

Once optimized parameters, optimal hydrogel (CurEpiβCD-Gel) was characterized in terms of viscosity, density, Curcumin content, gelation temperature, storage (G′) and loss moduli (G″), and in vitro Curcumin release and permeation in comparison with of the solution (CurEpiβCD-Sol).

Regarding gelling temperature, the effect of each of components of the optimized gel were analyzed. To facilitate interpretation, each component was added in sequence. Two approaches were taken for this assay: inversion test and G′–G″ crossover. The former is a simple assay with a precision of 2 °C but with difficulties to study cool temperatures. The latter is a more complex procedure that allows studies in cooler temperatures, but its results can vary with the frequency used in the protocol. For these reasons, both were used simultaneously.

As mentioned in Section 2.5.2, PluW behavior was in accordance with bibliography. When compared with PluWE, the inversion test showed no gelling temperature. However, tan (δ) showed a gelling temperature of 15.34 °C, which is considerably smaller than that of PluW. This indicated that ethanol is able to affect the micellation process of Pluronic^®^, augmenting it. As a consequence, the temperature dropped [31]. The addition of EpiβCD to PluWE increased the gelling temperature in comparison with PluWE, which is indicative of an interaction of the two components [49]. Interestingly, Epi-Gel and CurEpiβCD-Gel have similar gelation temperatures, indicating that Curcumin does not affect the structure of the system, although it can interact with Pluronic^®^ and EpiβCD at the same time. More about this interaction will be discussed later.

CurEpiβCD-Sol exhibited a rapid release for the first 8 h, reaching a maximum of 60% at 48 h. Afterward, Curcumin content declined, probably due to degradation in the dissolution medium. This medium is set at pH 7.4, in which Curcumin is partially ionized, as its first pK_a_ is 7–8, leading to its degradation [50]. On the other hand, the optimized formulation exhibited a considerably much slower release rate. This synergistic effect of CD complex and hydrogel has already been described by Sun et al., (2014), who demonstrated that the combination of CurHPβCD complex with a hydrogel resulted in a significantly slower release than the form CurHPβCD complex alone [51]. In comparison, CurEpiβCD-Gel provided a more sustained release than the formulation proposed by these authors. In addition, this combination also had an impact on the degradation process, as Curcumin content was maintained at 50% at the 72-h mark. This indicates a better stabilizing effect of the combination CD complex–hydrogel over CD complex alone in this medium.

In vitro permeation of Curcumin was also sustained over 72 h. In this case, the CurEpiβCD solution and hydrogel started with slow-rate permeation. Nonetheless, the solution reached its maximum at 24 h and, subsequently, started degrading, although not as markedly as in the release assay. EpiβCD is known for its ability to enhance drug permeation, as Gidwani et al., (2017) stated. In their studies, when tretinoin was combined with this polymeric CD and lipid nanostructures, the amount of drug permeated almost doubled [13]. In some permeation assays, raw Curcumin have demonstrated an extremely low permeation [5,52]. In comparison, this complex represents a huge improvement in Curcumin permeation. Regarding the hydrogel, an unexpected degradation occurred at 72 h, in which 15% of Curcumin was lost. This phenomenon will be further studied as Curcumin is more stable at acidic pH [50] and the experiment was performed in the absence of light. It should be noted that, even though the system degraded over time, Curcumin permeation was greater in the hydrogel than in the solution. Two main reasons behind this are the capability of Pluronic^®^ to form supramolecular structures with CD (more about this topic will be addressed later) and to the occlusive nature of the hydrogel.

The stability of the system was also tested. Three parameters were considered: pH, Curcumin antioxidant activity and Curcumin content. pH only decreased one unit after 21 days of experiment in both hydrogel systems (CurEpiβCD-Gel and Em-Gel), indicating a change in the structure of the gel. Considering Cur content, results showed that CurEpiβCD-Sol had better stability than the hydrogel. This can be attributed to gel composition, as it is the sole difference between both samples. pH could also contribute to Curcumin degradation, as it was higher in the hydrogel until day 21. However, degradation rate did not reduce when pH changed from 6 to 5, rendering this hypothesis as weak. Another hypothesis was that Pluronic^®^ or sodium hyaluronate are destabilizing the system. The poloxamer used in this work is a co-block polymer able to create supramolecular structures with CD. These structures were first introduced by Harada [53] and, since then, they have been well studied. Multiple of these studies include a variety of βCD derivatives [54,55,56,57], including polymeric CD such as EpiβCD [58], as well as EpiαCD [59]. In all cases, these supramolecular complexes formed stable systems with different drugs, enhancing their release and/or permeation profiles with the exception of DIMEB ((2,6-di-*O*-methyl)-βCD), which is capable of rupturing Pluronic^®^ micelles, a key part of this supramolecular structures [57]. On the other hand, sodium hyaluronate is a polysaccharide with high water binding capacity that can be useful in skin diseases treatment [60]. It has also been used in combination with CD, either in gel or to functionalize the CD [44,61,62,63], including combination with a poloxamer [44]. Again, there are no reports of sodium hyaluronate having a negative interaction with CD, including when it was combined with poloxamer.

From this investigation, we can deduce that neither Pluronic^®^ nor sodium hyaluronate destabilize CD system, either alone or in combination. In addition to this information, some types of Pluronic^®^ are able to stabilize Curcumin by entrapping the polyphenol in its lipophilic core. The conclusion that can be extracted is that Curcumin might not be suitable for these supramolecular structures in the conditions proposed, although they definitely benefit from it in terms of release and permeation.

Another positive effect of the CD was its capability to retain Curcumin antioxidant capability. On the one hand, the solution showed a consistent EC_50_ over the course of the experiment, with minimal changes. On the other hand, Curcumin in the hydrogel increased its antioxidant capability at days 15 and 21, to reduce it again.

Analyzing Curcumin content and antioxidant capability simultaneously, it is clear that, in both systems, the antioxidant effect cannot be attributed entirely to Curcumin. Many studies have identified Curcumin degradation products as ferulic acid, vanillin, ferulic aldehyde and feruloyl methane [3,63,64]. These products have in common the hydroxyl methoxyphenyl groups, that are key to the antioxidant activity of Curcumin [64]. Moreover, some authors even stated that the pharmacological effect of Curcumin can be attributed to its degradation products [65]. This would explain why, even when Curcumin presence in the system was diminishing, EC_50_ was kept unaltered (solution) or even increased (hydrogel).

Psoriasis is a chronic inflammatory skin disorder characterized by hyperproliferation of keratinocytes. Currently, treatments for this disease include topical and systemic agents, phototherapy and biologics. Regarding topical therapies, they are applied for mild forms of psoriasis, skin irritation being the most frequent adverse effect [24]. Therefore, new topical approaches for mild psoriasis with few side effects are required. In this regard, the use of natural products with anti-inflammatory properties is gaining considerable attention for psoriasis treatment. Previous in vitro and in vivo studies have reported the anti-psoriatic properties of Curcumin via suppression of pro-inflammatory cytokines, down-regulation of NF-kB and MAPK signaling pathways and upregulation of involucrin and filaggrin, proteins that regulate skin barrier function [66,67,68]. In addition, several formulations aimed to enhance bioavailability, solubility and physicochemical properties of Curcumin have been evaluated in different animal models of psoriasis, demonstrating an improvement of its anti-psoriatic effect [69,70,71]. In the present study, we demonstrated for the first time the anti-inflammatory properties of CurEpiβCD-Gel in an in vitro model of psoriasis, established using HaCaT keratinocytes treated with TNF-α. This cytokine is released by keratinocytes and plays a main role in the pathogenesis of psoriasis since it stimulates the production of other pro-inflammatory cytokines such as IL-6. This factor is a multifunctional cytokine that promotes epidermal keratinocyte hyperplasia and stimulates the differentiation of IL-17-producing T cells [72]. High plasma levels of this cytokine and its receptor have been detected in patients with psoriasis, demonstrating an important role in the pathogenesis of this disease [73]. For this reason, IL-6 inhibitors could be a promising strategy for inhibiting the inflammatory response in psoriasis. Our data evidenced that pre-treatment with CurEpiβCD-Gel significantly reduced the production of the pro-inflammatory cytokine IL-6 in TNF-α-stimulated HaCaT keratinocytes. These results are in agreement with previous studies that reported the capacity of Curcumin to reduce IL-6 levels in both HaCaT cells and psoriatic animals [65,67,73]. All these findings suggest that Curcumin, administered in a well-tolerated topical formulation, could be a novel and safe approach for the treatment of patients with mild psoriasis.

## 4. Materials and Methods

### 4.1. Materials

Soluble β-cyclodextrin-epichlorohydrin polymer (EpiβCD) MW 4500 cyclodextrin content (estimated 50–70%) residual β-cyclodextrin content max 1% was purchased from Cyclolab R&D Ltd. (Budapest, Hungary). Pluronic^®^ F-127 and high molecular weight chitosan were purchased from Sigma-Aldrich Co. (Barcelona, Spain). Sodium hyaluronate was purchased from Fagron Iberica (Barcelona, Spain). Carbopol^®^ 940 was purchased from Escuder (Barcelona, Spain). Solvents used for chromatographic analysis were HPLC quality. All other chemicals were of analytical degree.

### 4.2. Phase Solubility

An excess amount of Curcumin (10 mg) was added to 10 mL of phosphate buffer (pH 5) containing increasing amounts of EpiβCD. The concentration range for EpiβCD was 0–13.33 mM. The prepared samples in sealed glass containers protected from light were first sonicated 60 min in ultrasonic bath (Eurosonic 44 ultrasonic bath, Wilten Woltil de Meern, Utrecht, The Netherlands) and then magnetically stirred at constant temperature (25.0 ± 0.5 °C) until equilibrium (72 h). Aliquots were filtered (0.45 µm Millipore membrane filter, Darmstadt, Germany) and drug concentration was spectrophotometrically determined at 425 nm (Shimadzu 1601 UV-VIS spectrophotometer, Milan, Italy S.R.L). Preliminary studies showed that the presence of EpiβCD did not interfere with Curcumin absorbance at this wavelength. Each experiment was repeated at least 3 times (coefficient of variation, CV < 2%). The apparent stability constants of the complexes (Ks) were calculated from the straight portion of the phase-solubility diagrams using the following equation according to Higuchi and Connors [74]:(1)$$Ks=\frac{slope}{S_{0}\times \left(1-slope\right)}$$
where *S*_0_ is the aqueous solubility of Curcumin (mM) in the absence of CD.

### 4.3. Preparation of Interaction Products

Cur/EpiβCD 1:10 *w*/*w* physical mixture (PM) was prepared by adding accurately weighed components in a mortar and then mixing manually with a spatula for 13 min. Solid drug/CD interaction products were prepared by using different techniques: co-evaporated product (COE) was prepared by dissolving Curcumin in ethanol and adding a water solution of EpiβCD, then evaporating the solvent in a rotary evaporator at 70 °C and drying the resulting products in a vacuum desiccator with silica gel for 24 h to remove traces of solvents; for co-grinding (GR) product, the physical mixture was placed in a high-energy vibration micromill (MM200 Retsch GmbH, Haan, Germany) for 30 min at 24 Hz; kneading (KN) product was obtained pounding in a mortar the physical mixture, adding a minimum amount of a mixture water/ethanol 1:1 *v*/*v*, kneading thoroughly with a pestle to obtain a homogeneous slurry and continuing until the solvent was completely removed, then drying the products in a vacuum desiccator with silica gel for 24 h to remove traces of solvent; co-lyophilized product (COL) was prepared by dissolving Curcumin in the minimum amount of ethanol and adding a water solution of EpiβCD, then the solution was freeze-dried at −50 °C and 1.3·10^−2^ mm Hg. The 75–150 µm sieve granulometric fraction was then used for the following studies.

### 4.4. Differential Scanning Calorimetry (DSC)

Thermograms of the individual components or drug/EpiβCD products were obtained with a Mettler TA4000 calorimeter equipped with a DSC25 cell (Columbus, OH, USA). The samples, weighed with a Mettler M3 Microbalance (5–10 mg), were scanned in Al pans pierced with a perforated lid at 10 °C/min, from 30 to 150 °C, under static air.

### 4.5. Powder X-ray Diffractometry (PXRD)

A Bruker D8-advance (Billerica, MA, USA) X-ray diffractometer was employed for the diffraction patterns of drug and drug/EpiβCD binary systems. The parameters were: Cu Kα radiation, voltage 40 kV, current 55 mA and 2θ over a 5–45° range at a scan rate of 0.05°/s. A Sol-X^®^ solid-state Si (Li) was used as a detector, and C/Ni Goebel-Spiegel mirrors in the incident beam were used as a monochromator; 1.0 mm divergence, 0.2 scatter, and 0.1 for the receiving slits were used. All samples were examined at room temperature.

### 4.6. Fourier Transform Infrared Spectrometry (FT-IR)

FT-IR spectra were performed by a Shimadzu IRPrestige21 apparatus (Shimadzu Corporation, Kyoto, Japan) on KBr tablets. Spectra were recorded in the 400–4000 cm^−1^ range (10 scans, resolution 4 cm^−1^). Samples were examined after dispersion in Nujol.

### 4.7. Dissolution Rate Studies

In vitro dissolution studies of the prepared solid products (PM, GR, COE, COL, KN) and Curcumin were performed using a modified dispersed amount method [14]. Briefly, a sample amount corresponding to 10 mg of Curcumin was added to 20 mL of phosphate buffer (pH 5) at 37 °C and gently stirred (50 rpm) by a magnetic stirrer. At predetermined time intervals, aliquots of 3 mL were withdrawn with a syringe-filter (0.45 µm Millipore membrane filter) and immediately replaced with the same volume of fresh dissolution medium, thermostated at the same temperature. The drug amount in the samples was spectrophotometrically assayed as described previously. A correction was applied for the cumulative dilution. All experiments were repeated three times for each sample (CV < 2.5%).

### 4.8. Optimization of Gel Formulation

The main objective of the present work was to obtain a topical formulation of CurEpiβCD complex. For this, an experimental design–based screening study was realized in order to select the best composition of formulation. Four different polymers were evaluated: poloxamer in the form of Pluronic^®^ F-127, high molecular weight chitosan, carbomer in the form of Carbopol^®^ 940 and sodium hyaluronate. Due to its ability to jellify with temperature, Pluronic^®^ was used as the base for all gel formulations.

In order to optimize gel characteristics, an experimental design was performed using DOEpack 2000 as software (Dayton, OH, USA). The designated factors were: Curcumin concentration, type of polymer, Pluronic^®^/polymer ratio and polymer concentration. These factors and their levels are collected in Table 4. Taguchi L9 array was selected as design model. Responses analyzed were: percentage of drug released after 6 h, percentage of drug permeated at 6 h, apparent viscosity and pH of the formulation.

Gel preparation was as follows: Cur/EpiβCD binary system was dissolved in 3.6 mL of Pluronic^®^ 23.6% *w*/*v*. After 30 min, 1.4 mL of absolute ethanol was added, ensuring complete dissolution. Finally, when dissolution was completed, secondary polymer was added in the amount required by the array. The mixture was stirred and left to rest at 4 °C for 24 h. Afterwards, different responses were analyzed.

### 4.9. Characterization Studies

#### 4.9.1. Apparent Viscosity

Apparent viscosity was obtained with a rheometer (Discovery HR-3 hybrid rheometer, TA Instruments, New Castle, DE, USA). All rheological measurements were conducted by a parallel plate geometry (40 mm diameter) at 32 °C. Samples were analyzed monitoring the viscosity (η, Pa·s) as a function of the shear rate in the range of 10–1000 s^−1^. The gap of the assay was 400 μm. Apparent viscosity was selected at 62.65 s.

#### 4.9.2. Gelation Temperature

In order to comprehend the interaction between the components of the gel, the gelation temperature of the system was studied by inversion method and storage–loss moduli crossover.

For the inversion test, briefly, 1 mL of sample was placed in a glass tube and heated in thermostatic bath. Samples were heated at a rate of 1 °C/min from 25 to 50 °C. Every minute, the tube was turned for 30 s. The test was positive if gel sample stayed at the bottom of the tube. For ease of readability, data was displayed only at 25, 28, 32, 35, 41, 44, 47 and 50 °C, and assay was performed in duplicates. Formulation studied were: water solution of Pluronic^®^ F-127 17% *w*/*v* (PlusW), Pluronic^®^ F-127 17% *w*/*v* water/ethanol solution (PluWE), EpiβCD dissolved in Pluronic^®^ F-127 water/ethanol solution (Epi-Plu), optimized gel containing only EpiβCD (Epi-Gel) and Cur-EpiβCD-loaded optimized gel (CurEpiβCD-Gel). Samples were prepared following the procedure in Section 4.8.

For the storage–loss moduli crossover, moduli were analyzed following conditions in Section 4.9.3. Afterwards, loss tangent (tan (δ)) was calculated and the temperature or range of temperatures were tan (δ) was between 0.8 and 1.2 were selected as gelling temperature.

#### 4.9.3. Storage and Loss Moduli

To further analyze the optimized formula and the interaction between gel components, storage and loss moduli were calculated. For this matter, a rheometer (Discovery HR-3 hybrid rheometer, TA Instruments, New Castle, DE, USA) was employed. Samples were analyzed by monitoring storage (G′) and loss (G″) moduli as a function of temperature in the range of 15–65 °C. Frequency was kept at 1 Hz and oscillatory strain at 0.01%. Temperature was increased at 2 °C/min. Samples were analyzed in duplicates.

#### 4.9.4. In Vitro Release Studies

The release behavior of Curcumin from the hydrogel was evaluated by dialysis, as other studies have already performed [75]. Briefly, 1 g of each formulation was placed inside a dialysis bag (Spectra^®^/Por 12–14 kD MWCO membranes), previously rinsed with PBS pH 7.4/ethanol/Tween^®^ 80 mixture in ratio 74.5:25:0.5 (% *v*/*v*). The system was placed in 100 mL of the same medium under gentle stirring at room temperature. Samples (1 mL) were taken at scheduled times (0.5, 1, 1.5, 2, 3, 4, 5, 6, 8, 24, 48 and 72 h) and the same amount of fresh medium was added. Curcumin content was quantified using HPLC technique described in [33]. Each sample was analyzed in duplicate.

#### 4.9.5. In Vitro Permeation of Curcumin

To evaluate the amount of Curcumin able to penetrate skin, an in vitro permeation study was performed. In this case, a skin model was used instead of animal skin following Mura et al.’s (2014) protocol with slight modifications [76]. Cellulose membranes with a pore width of 0.2 µm was soaked in lauryl alcohol for 15 min at 58 °C. Afterwards, the excess of lauryl alcohol was dried with filter paper and weighed to ensure an increase in weight of 90–110%. Once dried, membranes were placed in Franz-type diffusion cells filled with PBS pH 5.5/ethanol/Tween^®^ 80 74.5:25:0.5 (% *v*/*v*) ensuring no air bubbles were present, mimicking skin conditions. Temperature was kept at 32 °C. Samples (0.5 mL) were withdrawn at scheduled times (0.5, 1, 1.5, 2, 3, 4, 5, 6, 8, 24, 48 and 72 h) and the same amount of fresh medium was added. Curcumin content was analyzed by HPLC as described in [33]. Duplicates of each sample were made.

#### 4.9.6. pH

This parameter was analyzed by dipping pH strips in the samples for 30 s. Results were read as by supplier instructions.

#### 4.9.7. Density

Hydrogel density was measured with a pycnometer, which was zeroed in a precision balance at 25 °C. Afterwards, gel formulation was added to the top avoiding air bubble formation and the system was closed with its corresponding cap, letting the excess gel to overflow. Finally, the pycnometer was cleaned and its weight measured. To obtain density, the following formula was applied:(2)$$D=\frac{Formulation\;weight-Pycnometer\;weight}{Pycnometer\;volume}$$

#### 4.9.8. Curcumin Extraction from Hydrogel

The amount of Curcumin that can be extracted from the optimized hydrogel was analyzed. In brief, an amount of hydrogel was accurately weighed, and a sufficient amount of chloroform was added. The mixture was vigorously stirred with the aid of a vortex mixer and the liquid was separated from the matrix. The sequence was repeated until chloroform was colorless. Afterwards, the extracted chloroform sample was centrifuged at 1000 rpm for 10 min at 4 °C to separate chloroform from any hydrogel residue and the organic phase was transferred to a round bottom flask to evaporate. Once all chloroform was evaporated, the Curcumin residue was dissolved in ethanol and analyzed by UV/Vis spectrophotometry using an Agilent 8453 UV-visible spectrophotometer (Agilent Technologies, Budapest, Hungary). A volume of 100 µL sample was diluted up to 5 mL with an acetonitrile/acetic acid 2% 1:1 *v*/*v* mixture, and absorbance was measured at 425 nm.

### 4.10. Stability Studies

Chemical stability of the optimized hydrogel and Curcumin embedded in gel was tested over the course of 84 days. pH, Curcumin content and Curcumin oxidation were analyzed. For comparison purposes, solution of CurEpiβCD complex and hydrogel without complex (Em-gel) were also considered. Analyses were made in duplicate. pH was measured as indicated in Section 4.9.6.

#### 4.10.1. Oxidation Study

Curcumin is an antioxidant agent capable of scavenging free radicals from both lipidic and aqueous phases [77]. In this study, the antioxidant capability of Curcumin was determined by using ABTS (2,2′-azino-bis(3-ethylbenzothiazoline-6-sulfonic acid) as oxidative agent, following the methodology previously reported by Pisoschi and Negulescu [78]. To create the oxidizing radical (ABTS•^+^), 2.98 mM of ABTS and 0.98 mM of K_2_S_2_O_8_ (potassium persulfate) were dissolved in purified water, forcing the loss of an electron from the nitrogen atom of ABTS. This dark blue solution is reduced by an antioxidant, losing its color in the process as the positively charged nitrogen is neutralized by a hydrogen from the antioxidant [77]. The decrease in the absorbance value was monitored, and Trolox (1.05 mg/mL in ethanol absolute) was chosen as standard antioxidant.

After obtaining the corresponding working solutions, serial dilutions were prepared into 96-well plates, adding 100 µL of working solutions and 100 µL of fresh ABTS, leaving to react for 6 min. In the case of Trolox, further dilution was needed, adding only 10 µL on working sample and 90 µL prior to fresh ABTS. Finally, absorbance at 734 nm was measured with a plate reader (Synergy HT Plate Reader, Winooski, VT, USA). The antioxidant activity was defined as EC_50_, which indicated the equivalent concentration of antioxidant needed to reduce the initial concentration of ABTS to 50%. Antioxidant capability was expressed as Trolox equivalent (EC_50_ compound/EC_50_Trolox) [33].

#### 4.10.2. Curcumin Content

In order to analyze Curcumin content, 100 µL samples were taken directly from the hydrogel and analyzed following the procedure described in Section 4.9.8.

### 4.11. Cell Studies

#### 4.11.1. Cell Viability Assay

Resazurin assay was used for determining the viability of HaCaT cells upon exposure to the optimized hydrogel. In this colorimetric assay, viable cells can reduce resazurin into the resorufin product, which is pink [79]. For this reason, not only the optimized gel was analyzed, but also each one of its components. In all cases, samples followed the same procedure described in Section 4.7, substituting hydrogel by water when needed. Therefore, formulations studied were: solution of CurEpiβCD complex in water (CurEpiβCD-Sol), optimized gel formulation containing only EpiβCD (EpiβCD-Gel), optimized gel formulation containing only Curcumin (Cur-Gel), unloaded optimized gel (Em-Gel) and CurEpiβCD-loaded optimized hydrogel (CurEpiβCD-Gel).

Firstly, HaCaT cells were seeded into 96-well plates in growth medium at 10^4^ cells/well for 24 h to ensure the adherence. Then, cells were incubated in a humidified atmosphere of 5% CO_2_ at 37 °C. After that, cells were treated with different formulations at final concentration range of 0.625–20 µM in ethanol 0.2% *v*/*v* and the cytotoxicity was measured after 48 h of incubation. Thereafter, cells were washed once with PBS pH 7.4, and 150 µL resazurin (20 µg/mL in medium) were added to each well. Plates were incubated for 4 h, and, finally, the absorbance was determined at 540 nm and 620 nm in a microplate spectrophotometer (Sinergy HT, Biotek^®^, Bad Friedrichshall, Germany). Cell viability was expressed as a percentage respect to the untreated cells.

#### 4.11.2. Determination of IL-6 Production

HaCaT cells were seeded in 6-well plates (2.5 × 10^5^ cells/well). After 24 h, cells were washed twice (PBS pH 7.4, 4 °C) and medium containing CurEpiβCD-Gel (2.5, 5 and 10 µM) was added for 24 h. Then, cells were stimulated with TNF-α (10 ng/mL) except for control group, for another 24 h. Thereafter, supernatant fluids were collected and stored at −80 °C until measurements. Commercial enzyme-linked immunosorbent assay (ELISA) kit (Diaclone GEN-PROBE, Besançon, France) was used to quantify IL-6 according to the manufacturer’s protocol.

#### 4.11.3. Statistical Analysis

All data in the figures are exhibited as arithmetic means with their standard error of means. Statistical analysis was carried out using GraphPad Prism version 5.00 software (GraphPad Software, Inc., San Diego, CA, USA). The Shapiro–Wilk test was used to verify the normality of the data. Student’s *t*-test was used to compare between the two control groups (control vs. TNF-α). Statistical differences between multiple groups were compared by one-way ANOVA followed by Bonferroni’s post hoc test. A *p*-value less than 0.05 was considered as statistically significant.

## 5. Conclusions

EpiβCD is a relatively new polymeric CD that is not yet widely studied. It is known for its capability to enhance solubility and permeation of poorly soluble molecules. For this reason, it was selected to improve Curcumin characteristics. In this study, this binary system was embedded in a hydrogel and its anti-inflammatory activity was tested.

CurEpiβCD binary system was characterized and the most suitable complexing method was determined. The results showed a high improved affinity between Curcumin and EpiβCD compared with other βCD and an impressive increase in dissolution rate with the GR method. Afterwards, a hydrogel system was optimized and characterized. Although the system demonstrated a reduced stability over time, permeation and release of Curcumin were highly improved. In addition, the optimized gel demonstrated anti-inflammatory activity. With these findings, we can conclude that CurEpiβCD binary system embedded in a Pluronic^®^/hyaluronate hydrogel is a promising treatment for skin inflammatory diseases.

## Figures and Tables

**Figure 1 ijms-22-13566-f001:**
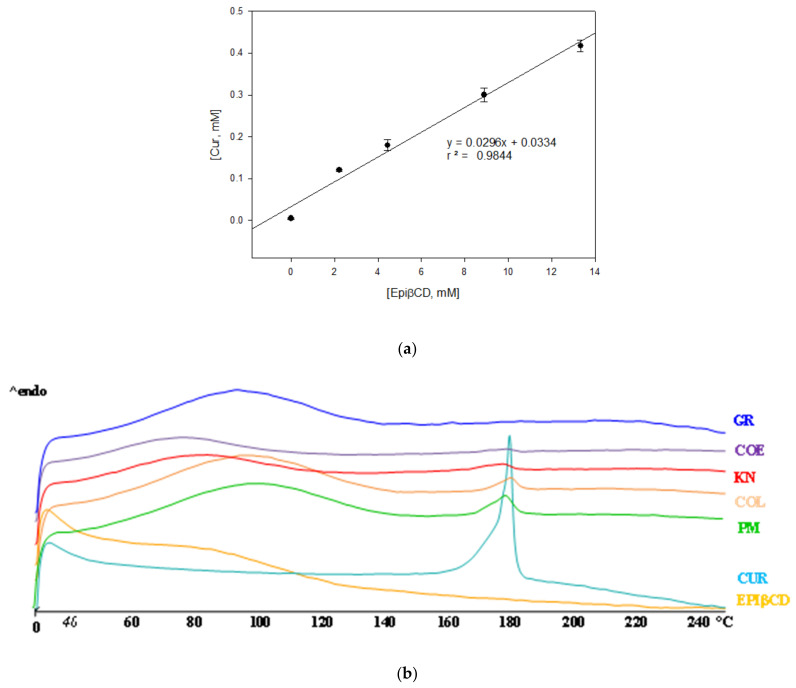
Phase solubility study and solid-state characterization of Cur/EpiβCD binary systems prepared with different methods. (**a**) Phase solubility study of Curcumin and EpiβCD. Thermograms by Differential Scanning Calorimetry, (**b**) X—ray diffraction spectra, (**c**) and FT—IR spectra, (**d**) Abbreviations: GR, co—grinding; COE, co—evaporation; KN, kneading; COL, co—lyophilization.

**Figure 2 ijms-22-13566-f002:**
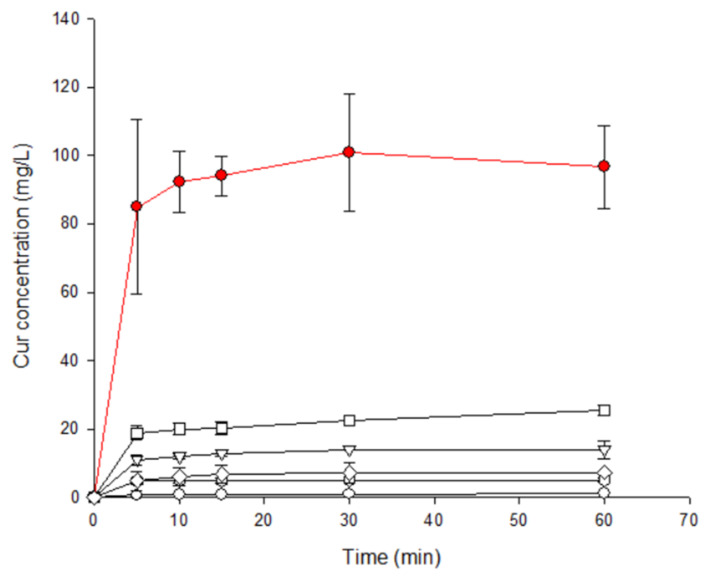
Dissolution rate of Cur/EpiβCD binary systems (GR 
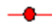
; COE 
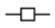
; KN 
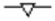
; COL 
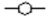
; PM 
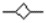
 ) compared with plain Cur 
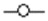
.

**Figure 3 ijms-22-13566-f003:**
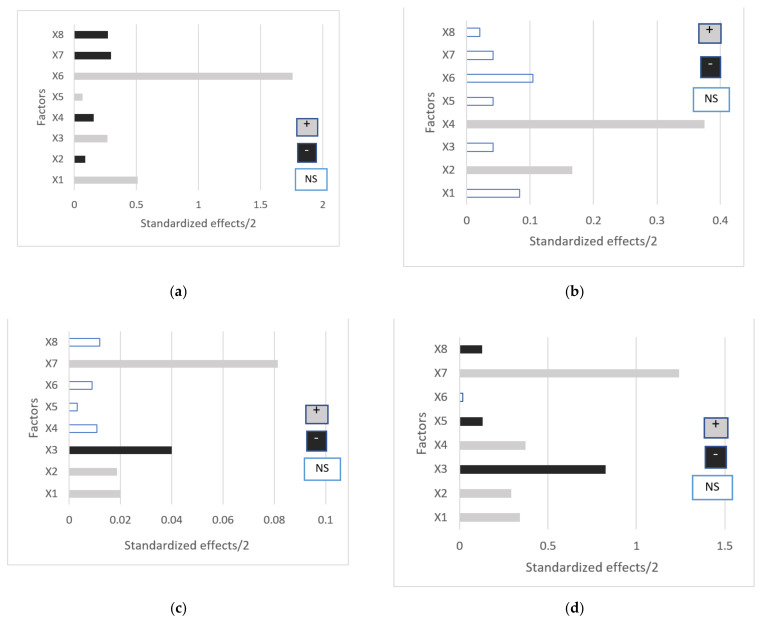
Pareto charts of half effects corresponding to: (**a**) gel viscosity; (**b**) pH; (**c**) in vitro Curcumin permeation; (**d**) in vitro Curcumin release. X1: [Pol] (−1 vs. +1); X2: Ratio (−1 vs. +1); X3: [Cur] (−1 vs. +1); X4: Tp (−1 vs. +1); X5: [Pol] (+1, −1 vs. 0). X6: Tp (−1, +1 vs. 0); X7: Ratio (−1, +1 vs. 0); X8: [Cur] (−1, +1 vs. 0).

**Figure 4 ijms-22-13566-f004:**
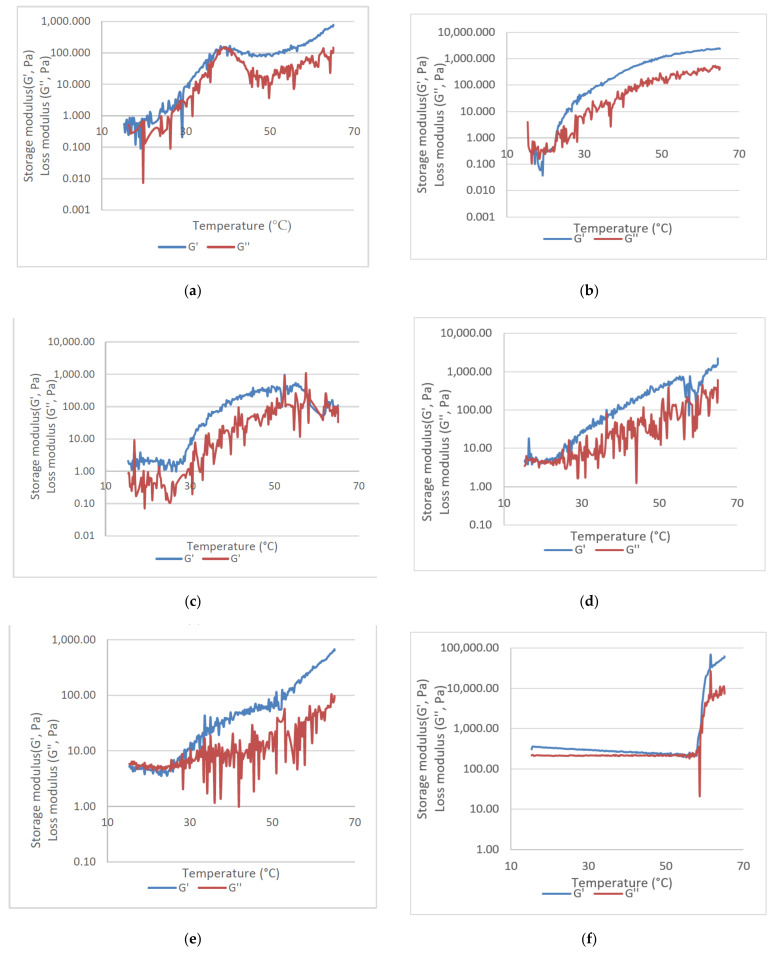
Storage (G′) and loss (G″) moduli for (**a**) PluW, (**b**) PluWE, (**c**) PluWE-Epi, (**d**) Epi-Gel, (**e**) CurEpiβCD-gel and (**f**) sodium hyaluronate.

**Figure 5 ijms-22-13566-f005:**
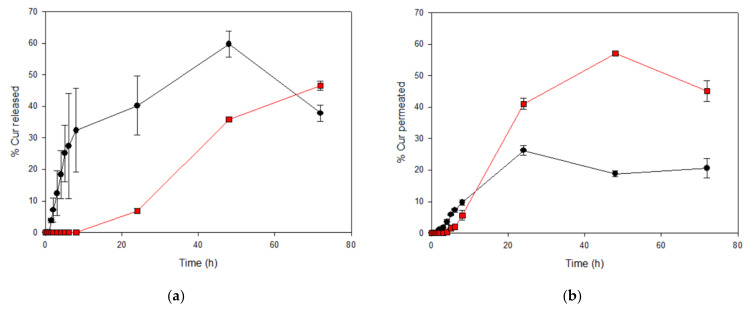
Percentage of Curcumin released (**a**) and permeated (**b**) from the optimized hydrogel (CurEpiβCD-Gel 
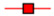
) and CurEpiβCD solution (CurEpiβCD-Sol 
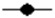
).

**Figure 6 ijms-22-13566-f006:**
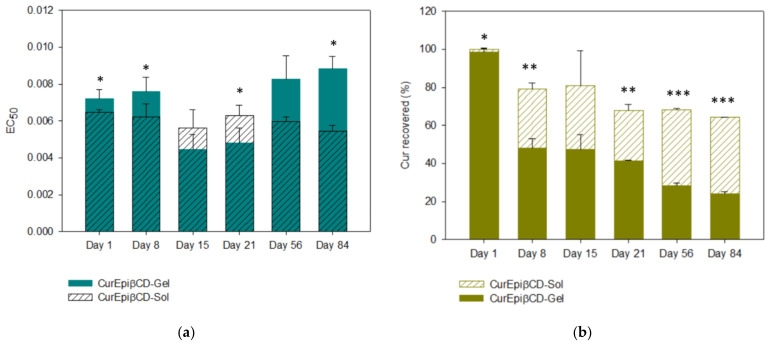
Antioxidant activity (**a**) and percentage of Curcumin recovered (**b**) from optimized hydrogel (CurEpiβCD-Gel) and solution (CurEpiβCD-Sol) over time. Data are expressed as the mean ± SD. The mean value of CurEpiβCD-Gel was significantly different compared with the control group, CurEpiβCD-Sol (* *p* < 0.05; ** *p* < 0.01; *** *p* < 0.001; Student’s *t*-test).

**Figure 7 ijms-22-13566-f007:**
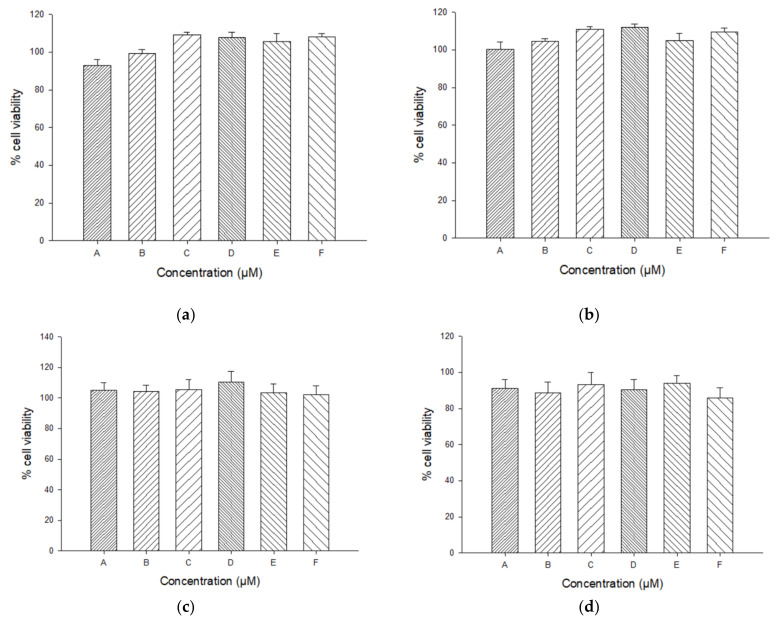
Cytotoxicity of the different samples: (**a**) Solution of CurEpiβCD complex (CurEpiβCD-Sol); (**b**) Optimized hydrogel without EpiβCD or Curcumin (Em-Gel); (**c**) Optimized hydrogel with EpiβCD only (EpiβCD-Gel); (**d**) Optimized hydrogel with Curcumin only (Cur-Gel); (**e**) Optimized CurEpiβCD-loaded hydrogel (CurEpiβCD-Gel).

**Figure 8 ijms-22-13566-f008:**
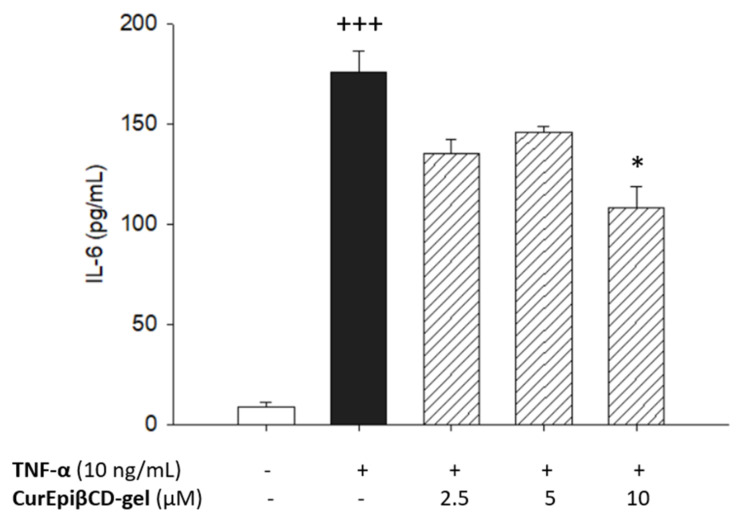
Effects of CurEpiβCD-Gel (2.5, 5 and 10 µM) on interleukin (IL)-6 levels in TNF-α-stimulated HaCaT keratinocytes. Production of IL-6 in supernatant was measured by ELISA assay. Results are representative of six independent experiments. Data are expressed as the mean ± SEM. The mean value was significantly different compared with the control group (+++ *p* < 0.001; Student’s *t*-test). Mean value was significantly different compared with the TNF-α group (* *p* < 0.05; one-way ANOVA followed by Bonferroni’s Multiple Comparison test).

**Table 1 ijms-22-13566-t001:** Experimental matrix for the Taguchi L9 array and results obtained. Results show the amount of Curcumin permeated (Permeation, %) and released (Release, %), gel apparent viscosity (Viscosity, Pa∙s) and pH. [Pol] represents the concentration of different polymers expressed in % *w*/*v*; Ratio represents the proportion in weight between Pluronic^®^ and the other polymer; [Cur] represents the Curcumin concentration expressed in mM; Tp represents the type of polymer as a non-numerical factor, choosing between chitosan (Chito), Carbopol^®^ (Carb) or sodium hyaluronate (Hyal).

Run	[Pol]	Ratio	[Cur]	Tp	Permeation ± SD	Release ± SD	Viscosity ± SD	pH ± SD
1	1	20:80	0.2	Chito	0.0963 ± 0.0044	1.25 ± 0.12	0.0698 ± 0.0017	4 ± 0
2	1	80:20	0.3	Carb	0.2070 ± 0.0089	3.28 ± 0.56	2.473 ± 0.045	4.2 ± 0.3
3	1	50:50	0.4	Hyal	0.075 ± 0.014	0.930 ± 0.098	0.114 ± 0.011	5 ± 0
4	2	20:80	0.3	Hyal	0.0678 ± 0.0044	0.9957 ± 0.0084	0.12 ± 0.11	4.7 ± 0.3
5	2	80:20	0.4	Chito	0.212 ± 0.011	2.452 ± 0.067	0.562 ± 0.077	4 ± 0
6	2	50:50	0.2	Carb	0.140 ± 0.019	2.25 ± 0.12	3.90 ± 0.17	4.5 ± 0
7	3	20:80	0.4	Carb	0.0491 ± 0.0071	0.618 ± 0.043	4.98 ± 0.12	4.2 ± 0.3
8	3	80:20	0.2	Hyal	0.339 ± 0.062	5.45 ± 0.23	0.0972 ± 0.0095	5 ± 0
9	3	50:50	0.3	Chito	0.1099 ± 0.00004	1.44 ± 0.13	0.64 ± 0.043	4.5 ± 0

**Table 2 ijms-22-13566-t002:** Summarized results of analysis of variance (ANOVA) of experimental results for apparent gel viscosity (Y_1_), pH (Y_2_), % Curcumin permeation at 6 h (Y_3_), % Curcumin release at 6h (Y_4_). X1: ([Pol] −1 vs. +1); X2: Ratio (−1 vs. +1); X3: [Cur] (−1 vs. +1); X4: Tp (−1 vs. +1); X5: [Pol] (+1, −1 vs. 0). X6: Tp (−1, +1 vs. 0); X7: Ratio (−1, +1 vs. 0); X8: [Cur] (−1, +1 vs. 0).

Factors	Viscosity (Y_1_)		pH (Y_2_)		Permeation (Y_3_)		Release (Y_4_)	
	F-Value	*p*-Value	F-Value	*p*-Value	F-Value	*p*-Value	F-Value	*p*-Value
X1	417.9	<0.001 *	2	0.191	9.1	0.015 *	28.9	<0.001 *
X2	12.1	0.007 *	8	0.020 *	7.991	0.020 *	21.28	0.001 *
X3	112.9	<0.001 *	0.5	0.497	36.47	<0.001 *	169.7	<0.001 *
X4	39.11	<0.001 *	40.5	<0.001 *	2.641	0.139	34.36	<0.001 *
X5	9.371	0.014 *	0.667	0.435	0.2954	0.600	5.714	0.041 *
X6	6612	<0.001 *	4.167	0.072	2.443	0.152	0.1107	0.747
X7	188.8	<0.001 *	0.6667	0.435	202.1	<0.001 *	509.8	<0.001 *
X8	157.1	<0.001 *	0.1667	0.693	4.303	0.068	5.326	0.46 *

* indicates which values are of statistical significance

**Table 3 ijms-22-13566-t003:** Gelling temperatures obtained by inversion test and loss tangent (tanδ) calculation. Samples are: Pluronic^®^ 17% *w*/*v* water solution (PluW), Pluronic 17% *w*/*v* water/ethanol solution (PluWE), PluWE containing EpiβCD (Plu-Epi), optimized gel containing only EpiβCD (Epi-Gel) and optimized gel loaded with CurEpiβCD GR binary system (CurEpiβCD-Gel). Inversion test was considered positive when gel remained at the bottom of the tube for 30 s after heating.

Sample	Inversion Test (°C)									Tan (δ)
	25	28	32	35	38	41	44	47	50	Range (°C)
Hyal	+	+	+	+	+	+	+	+	+	51.36–57.35
PluW	-	-	-	-	+	+	+	-	-	38.05–41.06
PluWE	-	-	-	-	-	-	-	-	-	15.34–23.09
PluWE-Epi	-	-	-	-	-	-	-	-	-	24
Epi-Gel	-	-	-	-	-	-	-	-	-	15.34–23.09
CurEpiβCD-Gel	-	-	-	-	-	-	-	-	-	15.34–27.77

**Table 4 ijms-22-13566-t004:** Factors (F1, F2, F3 and F4) and their levels selected for the screening study. Experiments were realized by using a Taguchi L9 array. (−1): low level; (0): middle level; (+1): high level.

Name	Factor	Level		
		−1	0	+1
F1	Cur concentration	0.2 mM	0.3 mM	0.4 mM
F2	Polymer type	Chitosan	Carbopol^®^	Hyaluronate
F3	Pluronic^®^/polymer ratio	20:80	80:20	50:50
F4	Polymer concentration	1% p/v	2% p/v	3% p/v

## Data Availability

Not applicable.

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
