# Peer review of "Preparation, Characterization and Evaluation of the Anti-Inflammatory Activity of Epichlorohydrin-β-Cyclodextrin/Curcumin Binary Systems Embedded in a Pluronic®/Hyaluronate Hydrogel"

_ijms, 2021, doi:10.3390/ijms222413566_

Round 1

Reviewer 1 Report

The manuscript submitted for review is very valuable and of a high level of merit. The compilation of so much experimental data and different research methods is impressive. This manuscript significantly exceeds the level of many articles published in this journal on the similar topics.

I certainly recommend the publication of this article. However, I would like to point out some minor issues that need to be corrected before accepting the manuscript. These are more of an editorial nature and the way the results are presented.

I have listed the points requiring improvement in the order they appear in the manuscript.

1) I would recommend avoiding the abbreviation "Cur" when the word curcumin is used alone. Of course, when using more complex formurlations, the abbreviation "Cur" is most welcome.
2) l. 95. Reference to the melting point of curcumin is missing.
3) The abbreviation "PM" first appears in l. 97, and is not explained until l. 100. An explanation of the abbreviation should appear after its first use. 
4) l. 107 and 110. Peak values in XRD are given without units. The correct unit is degree.
5) Fig. 1c and 1d. Units are missing on the horizontal axes.
6) Table 1. If the standard deviations are acting as uncertainties here then they should be corrected as follows. Uncertainty cannot have more significant digits than two. Uncertainty and result must have the same decimal expansion. 
7) l. 374. The atom "N" should be italicized here, while we write the configuration series designation in a much smaller font.
8) l. 442. Here similarly. The oxygen atom should be written in italics.
9) Experimental section. No information is given about the reagents, which incidentally were used a lot. The purity and supplier for each reagent should be given. 
10) l. 514-529. It is not stated what desiccants were used in the desiccator.
11) References. Of 69 total references, 11 direct to the authors' own papers. I believe that references numbered 13, 21, 33, 64, and 67-69 are completely unnecessary. It is better to replace them with works belonging to other researchers. 

Author Response

Point-by-point answers to Referee 1 comments

The manuscript submitted for review is very valuable and of a high level of merit. The compilation of so much experimental data and different research methods is impressive. This manuscript significantly exceeds the level of many articles published in this journal on the similar topics.

I certainly recommend the publication of this article. However, I would like to point out some minor issues that need to be corrected before accepting the manuscript. These are more of an editorial nature and the way the results are presented.

I have listed the points requiring improvement in the order they appear in the manuscript.

Comment 1: I would recommend avoiding the abbreviation "Cur" when the word curcumin is used alone. Of course, when using more complex formurlations, the abbreviation "Cur" is most welcome.

Answer to comment 1: As the Referee suggested, the use of Cur as an abbreviation of Curcumin was eliminated when curcumin is used alone

Comment 2:
L. 95. Reference to the melting point of curcumin is missing.

Answer to comment 2: A the reviewer suggested, a reference to curcumin melting point was added. The paragraph changed as follows:

In order to evaluate the effect of the preparation techniques, different bi-nary systems of Cur with EpiβCD were prepared and submitted to the DSC analysis. As shown in Figure 1b, Cur presents a melting peak at 179.72 °C with a melting enthalpy of 95.42 J/g, which is in accordance with bibliography [26]

Ref 26: Yadav, V.R.; Suresh, S.; Devi, K.; Yadav, S. Effect of Cyclodextrin Complexation of Curcumin on Its Solubility and Antiangiogenic and Anti-Inflammatory Activity in Rat Colitis Model. AAPS PharmSciTech 2009, 10, 752–762, doi:10.1208/s12249-009-9264-8

Comment 3: The abbreviation "PM" first appears in l. 97, and is not explained until l. 100. An explanation of the abbreviation should appear after its first use. 

Answer to comment 3: The abbreviation was clarified in line 97 instead of line 100. Paragraph changed to:

Cur melting peak was almost unchanged in the physical mixture (PM) with EpiβCD, indicating the absence of solid-state interactions between the compo-nents. On the contrary, the drug melting peak intensity gradually decreased as a function of the technique used, with the trend PM = kneading (KN) < coly-ophilizated (COL) < coevaporated (COE), and it completely disappeared in the co-grinding (GR). This indicated that co-grinding procedure was the most successful to induce effective solid-state interactions between the components, resulting in complete Cur amorphization and/or complexation:

Comment 4:
l. 107 and 110. Peak values in XRD are given without units. The correct unit is degree.

Answer to comment 4: Units were added to the peak values, changing the sentence as follows:

Diffraction patterns of the PM, KN and COL products almost corresponded to the superimposition of those of the plain components, where the peaks at 9, 15 and 18° were still present, confirming a limited interaction between the com-ponents

Comment 5: Fig. 1c and 1d. Units are missing on the horizontal axes.

Answer to comment 5: Units were added in these axes and quality of the figures were also improved. To see changes in the manuscript, referee is refered to the manuscript

Comment 6: Table 1. If the standard deviations are acting as uncertainties here then they should be corrected as follows. Uncertainty cannot have more significant digits than two. Uncertainty and result must have the same decimal expansion. 

Answer to comment 6: As the referee pointed out, standard deviation in Table 1 were acting as uncertainties. Data were corrected to follow proper form. Table 1 was changed to:

Run

[Pol]

Ratio

[Cur]

Tp

Permeation ± SD

Release ± SD

Viscosity ± SD

pH ± SD

1

1

20:80

0.2

Chito

0.0963±0.0044

1.25±0.12

0.0698±0.0017

4±0

2

1

80:20

0.3

Carb

0.2070±0.0089

3.28±0.56

2.473±0.045

4.2±0.3

3

1

50:50

0.4

Hyal

0.075±0.014

0.930±0.098

0.114±0.011

5±0

4

2

20:80

0.3

Hyal

0.0678±0.0044

0.9957±0.0084

0.12±0.11

4.7±0.3

5

2

80:20

0.4

Chito

0.212±0.011

2.452±0.067

0.562±0.077

4±0

6

2

50:50

0.2

Carb

0.140±0.019

2.25±0.12

3.90±0.17

4.5±0

7

3

20:80

0.4

Carb

0.0491±0.0071

0.618±0.043

4.98±0.12

4.2±0.3

8

3

80:20

0.2

Hyal

0.339±0.062

5.45±0.23

0.0972±0.0095

5±0

9

3

50:50

0.3

Chito

0.1099±0.00004

1.44±0.13

0.64±0.043

4.5±0

Comments 7 and 8: L. 374. The atom "N" should be italicized here, while we write the configuration series designation in a much smaller font. L. 442. Here similarly. The oxygen atom should be written in italics.

Answers to comment 7 and 8: Changes in the font of the oxygen and nitrogen atoms were performed, resulting in the following changes:

Chitosan is a polysaccharide composed of N-acetyl-D-glucosamine and D-glucosamine units linked by 1-4-β-glycosidic bonds

In all cases, these supramolecular complexes formed stable systems with different drugs, enhancing their release and/or permeation profiles with the exception of DIMEB ((2,6-di-O-methyl)-βCD), that is capable of rupturing Pluronic® micelles, a key part of this supramolecular structures

Comment 9: Experimental section. No information is given about the reagents, which incidentally were used a lot. The purity and supplier for each reagent should be given. 

Answer to comment 9: A material section was added indicating the reagents use and purchasing distributors. For this purpose, a new section was created (section 4.1) and numeration of the material section was corrected accordingly. It reads as follows:

4.1. Materials

Soluble β-cyclodextrin-epichlorohydrin polymer (EpiβCD) MW 4500 cy-clodextrin content (estimated 50 - 70%) residual β-cyclodextrin content max 1% was purchased from Cyclolab R&D Ltd (Hungary). Pluronic® F-127 and high molecular weight chitosan were purchased from Sigma-Aldrich Co (Bar-celona, Spain). Sodium hyaluronate was purchased from Fagron Iberica (Bar-celona, Spain). Carbopol® 940 was purchased from Escuder (Barcelona, Spain). Solvents used for chromatographic analysis were HPLC quality. All other chemicals were of analytical degree.

Comment 10: L. 514-529. It is not stated what desiccants were used in the desiccator.

Answer to comment 10: The type of desiccant employed was added to the text, resulting as follows:

4.3. Preparation of interaction products

Cur:EpiβCD 1:10 w/w physical mixture (PM) was prepared by adding accurately weighed components in a mortar and then mixing manually with a spatula for 13 minutes. Solid drug/CD interaction products were prepared by using different techniques: co-evaporated product (COE) was prepared by dissolving Cur in ethanol and adding a water solution of EpiβCD, then evaporating the solvent in a rotary evaporator at 70 °C and drying the resulting products in a vacuum desiccator with silica gel for 24 h to remove traces of solvents; for co-grinding (GR) product, the physical mixture was placed in a high-energy vibration micromill (MM200 Retsch GmbH, Germany) for 30 minutes at 24 Hz; kneading (KN) product was obtained pounding in a mortar the physical mixture, adding a minimum amount of a mixture water:ethanol 1:1 v/v, kneading thoroughly with a pestle to obtain a homogeneous slurry and continuing until the solvent was completely removed, then drying the products in a vacuum desiccator with silica gel for 24 h to remove traces of solvent; co-lyophilized product (COL) was prepared by dissolving Cur in the minimum amount of ethanol and adding a water solution of EpiβCD, then the solution was freeze-dried at -50 °C and 1.3·10-2 mm Hg. The 75–150 µm sieve granulometric fraction was then used for the following studies.

Comment 11: References. Of 69 total references, 11 direct to the authors' own papers. I believe that references numbered 13, 21, 33, 64, and 67-69 are completely unnecessary. It is better to replace them with works belonging to other researchers. 

Answer to comment 11: As the referee pointed out, 11 of the 69 references provided in the original manuscript corresponded to our own papers. We carefully studied these references and decided to eliminate, change or maintain depending on the area (for better understanding, new reference numbers will be provided in brackets):

Ref 13: This reference was maintained as it serves as background to protocol. As the problem was that too many auto-references were made, ref 14 Jug, M.; Maestrelli, F.; Mura, P. Native and Polymeric β-Cyclodextrins in Performance Improvement of Chitosan Films Aimed for Buccal Delivery of Poorly Soluble Drugs. J. Incl. Phenom. Macrocycl. Chem. 2012, 74, 87–97, doi:10.1007/s10847-011-0086-4  was change for Gidwani, B.; Jaiswal, P.; Vyas, A. Formulation and Evaluation of Gel Containing Nanostructured Lipid Carriers of Tretinoin–Epi-β-CD Binary Complex for Topical Delivery. J. Incl. Phenom. Macrocycl. Chem. 2017, 89, 315–323, doi:10.1007/s10847-017-0747-z (ref 14)

Ref 21: this reference was changed to “Gidwani, B.; Jaiswal, P.; Vyas, A. Formulation and Evaluation of Gel Containing Nanostructured Lipid Carriers of Tretinoin–Epi-β-CD Binary Complex for Topical Delivery. J. Incl. Phenom. Macrocycl. Chem. 2017, 89, 315–323, doi:10.1007/s10847-017-0747-z.” (Ref 14)

Ref 33 (Ref 44): This reference remained in the text as it provided background for discussion section. Specifically, the absence of negative interaction between Pluronic® F-127 and sodium hyaluronate. In the text:

It has also been used in combination with CD, either in gel or functionalizing the CD [44,59–61], including combination with a poloxamer [44].

Ref 64 (Ref 76): This reference remained in the text as it provided background for a protocol of artificial membrane. Many skin models have been developed over the years, making more necessary to reference previous work.

Ref 67 and 69: These references were eliminated as they did not provide more information than the text is already providing.

Ref 68 (Ref 79): This reference was change to  Präbst, K.; Engelhardt, H.; Ringgeler, S.; Hübner, H. Basic Colorimetric Proliferation Assays: MTT, WST, and Resaz-urin. In Methods in Molecular Biology; Springer New York, 2017; Vol. 1601, pp. 1–17 ISBN 1064-3745.

Reviewer 2 Report

This paper describes the production of epichlorohydrin-β-cyclodextrin/curcumin inclusion complex embedded Pluronic®/hyaluronate hydrogels to be employed for the release of a natural anti-inflammatory agent (i.e., curcumin). Due to the complexation ability of cyclodextrin with curcumin, which was used for an inflammatory drug, inclusion complexes were formed between the epichlorohydrin-b-CD and curcumin. The experiments were done carefully. However, there are some critical issues listed below. Please consider revising the manuscript according to the following comments/suggestions.

Line 19, typo! Epichlorohydrine-β-CD should be Epichlorohydrin-β-CD

Line 63, Can the authors cite some works on ECH-β-CD in bio-related applications, particularly in vivo use?

Line 106, figure 1c should be Figure 1c

Line 110 peaks at 9, 15, and 18 2Ө should be 9, 15 and 18o

Line 114, 1627-1602 cm-1, the longer dash should be used when referring to the range

Fig. 1 was poorly presented. The authors should revise the figure, and accordingly, the caption should be revised.

Figure 5. the font size should be increased for better readability.

Figures 6 & 7 these figures can be a standard format for other figures in the manuscript. Consider revising the other figures for font size, line thickness, and formatting accordingly.

Ref 2, the page numbers are missing (Volume 83, September 2015, Pages 111-124)

Ref 34, the article number is missing (AAPS PharmSciTech volume 22, Article number: 103 (2021))

Ref 65, issue and page numbers are missing (Analyst, 2002, 127, 183-198)

Line 495, Materials and Methods, should start with the material section, where the details of reagents should be given. What are the suppliers and the purity degrees, if possible? What is the molecular weight of Epi-b-CD?

Line 590-591, cone plate or parallel plate geometry? If the cone plate geometry was used, what was the angle?

The resulting gels have to be characterized. What are the gelling temperatures? Does it change according to the additives added to the gelling solutions? Elastic and loss moduli can be followed as a function of temperature.. Likewise, frequency and strain sweep experiments can be done to characterize the mechanical properties of the gels.

Photos of the gels should be given as well. They can be incorporated into the ESI.

Fig. 1b, it is impossible to read the x-axis! All the figures should be prepared in the same or similar graph style for better readability.

Fig. 1d, the narrower range (e.g., 1400-1700 cm-1) can be shown to emphasize the changes for the readers.

Fig. 2 shows that there is a burst release for almost all cases, followed by a very gradual or no increase in the release profiles, which are not desired for sustained release of the embedded drug molecules to the targeted media or tissue. Can the authors comment on this in the text?

Author Response

Point-by-point answers to Referee 2 comments

This paper describes the production of epichlorohydrin-β-cyclodextrin/curcumin inclusion complex embedded Pluronic®/hyaluronate hydrogels to be employed for the release of a natural anti-inflammatory agent (i.e., curcumin). Due to the complexation ability of cyclodextrin with curcumin, which was used for an inflammatory drug, inclusion complexes were formed between the epichlorohydrin-b-CD and curcumin. The experiments were done carefully. However, there are some critical issues listed below. Please consider revising the manuscript according to the following comments/suggestions.

Comment 1: Line 19, typo! Epichlorohydrine-β-CD should be Epichlorohydrin-β-CD

Answer to comment 1: The typo was addressed and changed accordinglo (Line19)

Comment 2: Line 63, Can the authors cite some works on ECH-β-CD in bio-related applications, particularly in vivo use?

Answer to comment 2: Two new references were added to provide background requested. Text was changed to:

Among the derivatives, soluble epichlorohydrin polymer derivative of β-CD  (EpiβCD) showed a great ability on improving drug solubility [13,14] and oral bioavailability [15,16]  after its incorporation in drug formulation

Ref 16: Gidwani, B., & Vyas, A. (2017). Pharmacokinetic study of solid-lipid-nanoparticles of altretamine complexed epichlorohydrin-β-cyclodextrin for enhanced solubility and oral bioavailability. International Journal of Biological Macromolecules, 101, 24–31. https://doi.org/10.1016/J.IJBIOMAC.2017.03.047

Ref 15: Nie S, Zhang S, Pan W, Liu Y. In vitro and in vivo studies on the complexes of glipizide with water-soluble β-cyclodextrin-epichlorohydrin polymers. Drug Dev Ind Pharm. 2011 May;37(5):606-12. doi: 10.3109/03639045.2010.533277. PMID: 21469949.

Comment 3: Line 106, figure 1c should be Figure 1c

Answer to comment 3: Typing error was corrected (Line 118).

Comment 4: Line 110 peaks at 9, 15, and 18 2Ө should be 9, 15 and 18o

Answer to comment 4: Units of the peaks were corrected and 2Ө was changed to degree (Line 122-123)

Comment 5: Line 114, 1627-1602 cm-1, the longer dash should be used when referring to the range

Answer to comment 5: A longer dash was employed instead of the short dash in line 127.

Comment 6: Fig. 1 was poorly presented. The authors should revise the figure, and accordingly, the caption should be revised.

Answer to comment 6: Figure 1 was revised. Figures 1a and 1b were separated in different lines to allow for a better reading of the axis. Figure 1c had color changed for better understanding, legend positioned above the graph and units indicated in the corresponding axis. Figure 1d was also changed, adding corresponding units to the axis and improving legend for simple understanding. Also, meaning of the abbreviations was added to the title of the figure to facilitate comprehension. 

Comment 7: Figure 5. the font size should be increased for better readability.

Answer to comment 7: Font of Figure 5 was increased as suggested.

Comment 8: Figures 6 & 7 these figures can be a standard format for other figures in the manuscript. Consider revising the other figures for font size, line thickness, and formatting accordingly.

Answer to comment 8: Figures were changed accordingly to the suggestions by referee2

Comments 9, 10 and 11: Ref 2, the page numbers are missing (Volume 83, September 2015, Pages 111-124). Ref 34, the article number is missing (AAPS PharmSciTech volume 22, Article number: 103 (2021)). Ref 65, issue and page numbers are missing (Analyst, 2002, 127, 183-198)

Answer to comments 9, 10 and 11: Bibliography error sin references 2, 34 and 65 were corrected. See references 2, 43 and 76 in the corrected manuscript.

Comment 11: Line 495, Materials and Methods, should start with the material section, where the details of reagents should be given. What are the suppliers and the purity degrees, if possible? What is the molecular weight of Epi-b-CD?

Answer to comment 11: As suggested by the referee, a material section was added to the Material an Methods chapter. The degree of purity and supplier of the reagents were added, as well as the molecular weight of Epi-β-CD (section 4.1). The test changed has follow:

4.1. Materials

Soluble β-cyclodextrin-epichlorohydrin polymer (EpiβCD) MW 4500 cy-clodextrin content (estimated 50 - 70%) residual β-cyclodextrin content max 1% was purchased from Cyclolab R&D Ltd (Hungary). Pluronic® F-127 and high molecular weight chitosan were purchased from Sigma-Aldrich Co (Bar-celona, Spain). Sodium hyaluronate was purchased from Fagron Iberica (Bar-celona, Spain). Carbopol® 940 was purchased from Escuder (Barcelona, Spain). Solvents used for chromatographic analysis were HPLC quality. All other chemicals were of analytical degree.

Comment 12: Line 590-591, cone plate or parallel plate geometry? If the cone plate geometry was used, what was the angle?

Answer to comment 12: The type of plate of the rheometer was added to the text, that changed to:

Apparent viscosity was obtained with a rheometer (Discovery HR-3 hybrid rheometer, TA Instruments, Delaware, USA). All rheological measurements were conducted by a parallel plate geometry (40 mm diameter) at 32 °C. Samples were analyzed monitoring the viscosity (η,Pa·s) as a function of the shear rate in the range of 10–1000 s−1. The gap of the assay was 400 μm. Apparent viscosity was selected at 62.65 s.

Comment 13: The resulting gels have to be characterized. What are the gelling temperatures? Does it change according to the additives added to the gelling solutions? Elastic and loss moduli can be followed as a function of temperature. Likewise, frequency and strain sweep experiments can be done to characterize the mechanical properties of the gels. Photos of the gels should be given as well. They can be incorporated into the ESI.

Answer to comment 13: As the referee proposed, storage and loss moduli and gelling temperature were analyzed. However, frequency and strain sweeps were not performed as the main objective of the manuscript is to investigate the effect of EpiβCD over curcumin. To accommodate the new assays, sections 4.9.2 and 4.9.3 were added to the Material and Methods chapter. Subsequently changes in the Results (section 2.5.2) and Discussion (lines 484-505) chapters were also performed. Additionally, some photos were taken to add as content for the ESI.

Comments 14 and 15: Fig. 1b, it is impossible to read the x-axis! All the figures should be prepared in the same or similar graph style for better readability. Fig. 1d, the narrower range (e.g., 1400-1700 cm-1) can be shown to emphasize the changes for the readers.

Answer to comments 14 and 15: As mentioned before, Figure 1b was changed for better readability. Additionally, from Figure 1d was created an emphasize figure from the range 1400-900 cm-1 for better understanding of the image. However, due to the magnitude of it, we consider a better option to display it in the ESI. This figure has already been referenced in the main manuscript, see line 128.

Comment 16: Fig. 2 shows that there is a burst release for almost all cases, followed by a very gradual or no increase in the release profiles, which are not desired for sustained release of the embedded drug molecules to the targeted media or tissue. Can the authors comment on this in the text?

Answer to comment 16: Figure 2 shows the dissolution rate of curcumin from the different binary systems with EpiβCD (and not its release from the final formulation). Curcumin is an almost insoluble drug, and the preparation of binary systems with EpiβCD (particularly by cogrinding, GR) lead to a significant improvement of its solubility and dissolution rate that’s our goal. For a topical application a sustained release is given by the gel formulation and by the skin barrier that allow, respectively, the release and the permeation only of the drug fraction in solution.

Reviewer 3 Report

Authors developed a Pluronic® 23 F-127/hyaluronate hydrogel loaded with a complex prepared between Epichlo- 18 rohydrine-β-CD (EpiβCD) and curcumin for the skin application of this active ingredient.

The topic is interesting but in my opinion in this formi is not acceptable for publication. 

Introduction. Authors did not reported literature data about the combination of cyclodestrin complexes with hydrogel. This part must be intorduced.

Line 63. Authors state “At the moment, there are no studies of complexation of Cur with EpiβCD”. It is not correct. See the article “Characterization of Curcumin/Cyclodextrin Polymer Inclusion Complex and Investigation on Its Antioxidant and Antiproliferative Activities” Molecules. 2018 May; 23(5): 1179.

Line 79. Why the authors performed the experiment at pH 5? Generally pH 5,5 is used. Please justify and insert reference to justify this choice.

Figure 1. The resolution is low and the figure too confusing. Panel b: x axis is not visible. Panel c: the graphs must be replaced. The title of x axis must be written correctly. Panel d. x axis title is missing. More peaks must be identified, 1583, 1628, 1602 unit??? The identification of each sample is pretty complicate. The caption must be improved introducing the corrispondence between the abbreviation and the sample name.

Lines 146-149: “the poloxamer used is not able to reticulate under 17 % w/v while sodium hyaluronate, chitosan and Carbopol® are able to jellify at much lower concentration” this must be supported by references. Why did you selected these polymers?

Line 324. What is the value of Cur solubility in water?

Line 413. Permeation studies. Why did you evaluate the permeation during 72 h? It is not reasonable to think that this formulation can stay on skin surface for a long time. Many studies document the non correspondence between the use of artificial and natural membranes. This experiment could not reproduce the real use conditions.

Line 553. modified dispersed amount method, a reference is necessary? Why the authors did not used official methods reported in Pharmacopoeia?

Par. 4.8.2. The release behavior of Cur from the hydrogel was evaluated by dialysis. This method is not acceptable to evaluate the release from a semisolid formulation. Official methods must be used for example Franz diffusion cell (USP).

Line 597. “PBS pH 7.4”. Why did you used this pH? It is not applicable for formulations intended to be applied on skin.

Line 582. “After the binary system was as dissolved as possible” This phrase is not scientific

The english must be revised

Author Response

Point-by-point answers to Referee 3 comments

Authors developed a Pluronic® 23 F-127/hyaluronate hydrogel loaded with a complex prepared between Epichlorohydrine-β-CD (EpiβCD) and curcumin for the skin application of this active ingredient.

The topic is interesting but in my opinion in this formi is not acceptable for publication. 

Comment 1: Introduction. Authors did not reported literature data about the combination of cyclodestrin complexes with hydrogel. This part must be intorduced.

Answer to comment 1: As the referee required, data regarding the incorporation of the cyclodextrin to hydrogels was added. The text changed as follows:

Several recent studies have been reported about the combination of cyclodextrins complexes and hydrogel [19–21] also for curcumin administration [22].

Ref 19: Cui, H., Wang, Y., Li, C., Chen, X., & Lin, L. (2021). Antibacterial efficacy of Satureja montana L. essential oil encapsulated in methyl-β-cyclodextrin/soy soluble polysaccharide hydrogel and its assessment as meat preservative. LWT, 152, 112427. https://doi.org/10.1016/J.LWT.2021.112427

Ref 20: Garg, A., Ahmad, J., & Hassan, M. Z. (2021). Inclusion complex of thymol and hydroxypropyl-β-cyclodextrin (HP-β-CD) in polymeric hydrogel for topical application: Physicochemical characterization, molecular docking, and stability evaluation. Journal of Drug Delivery Science and Technology, 64, 102609. https://doi.org/10.1016/J.JDDST.2021.102609

Ref 21: Shabkhiz, M. A., Khalil Pirouzifard, M., Pirsa, S., & Mahdavinia, G. R. (2021). Alginate hydrogel beads containing Thymus daenensis essential oils/Glycyrrhizic acid loaded in β-cyclodextrin. Investigation of structural, antioxidant/antimicrobial properties and release assessment. Journal of Molecular Liquids, 344, 117738. https://doi.org/10.1016/J.MOLLIQ.2021.117738

Ref 22:Torchio, A., Cassino, C., Lavella, M., Gallina, A., Stefani, A., Boffito, M., & Ciardelli, G. (2021). Injectable supramolecular hydrogels based on custom-made poly(ether urethane)s and α-cyclodextrins as efficient delivery vehicles of curcumin. Materials Science and Engineering: C, 127, 112194. https://doi.org/10.1016/J.MSEC.2021.112194)

Comment 2: Line 63. Authors state “At the moment, there are no studies of complexation of Cur with EpiβCD”. It is not correct. See the article “Characterization of Curcumin/Cyclodextrin Polymer Inclusion Complex and Investigation on Its Antioxidant and Antiproliferative Activities” Molecules. 2018 May; 23(5): 1179.

Answer to comment 2: We thank the Reviewer for his comment. The sentence has been modified as follows:

A recent study showed the enhanced antioxidant and antiproliferative activity of curcumin obtained by complexation with a purposely synthesized EpiβCD [17].

Ref 17: Chen J, Qin X, Zhong S, Chen S, Su W, Liu Y. Characterization of Curcumin/Cyclodextrin Polymer Inclusion Complex and Investigation on Its Antioxidant and Antiproliferative Activities. Molecules. 2018 May 15;23(5):1179. doi: 10.3390/molecules23051179. PMID: 29762477; PMCID: PMC6100345.

Comment 3: Line 79. Why the authors performed the experiment at pH 5? Generally pH 5,5 is used. Please justify and insert reference to justify this choice.

Answer to comment 3: pH 5 was chosen as a compromise between topical pH and acid pH which allows for greater stability. This was also clarified in the text for better comprehension of the manuscript. The text changed as follow:

This pH was selected as a compromise between Cur stability and skin pH [25].

Ref 25: Jain, B. (2017). A spectroscopic study on stability of curcumin as a function of pH in silica nanoformulations, liposome and serum protein. Journal of Molecular Structure, 1130, 194–198. https://doi.org/10.1016/J.MOLSTRUC.2016.10.014

Comment 4: Figure 1. The resolution is low and the figure too confusing. Panel b: x axis is not visible. Panel c: the graphs must be replaced. The title of x axis must be written correctly. Panel d. x axis title is missing. More peaks must be identified, 1583, 1628, 1602 unit??? The identification of each sample is pretty complicate. The caption must be improved introducing the corrispondence between the abbreviation and the sample name.

Answer to comment 4: As the referee pointed out, there were missing information and unreadable information in this figure. For this reason, it was change in the following manner:

Figure 1a and1b were separated in two lines to allow for a better expansion of the figures.

Figure 1b font was augmented and legend was improved with more contrasting colors. Also, the direction of endothermic effect was added.

Figure 1c and d had added corresponding units to the axis. Legends were amplified and placed above the graph, units were indicated in the corresponding axis and graphs were improved, allowing for better interpretation.

In Figure 1d, arrows were used to better correlation with legend and units were added to the corresponding axis.

Additionally, meaning of the abbreviations was added to the title of the figure to facilitate comprenhention.

Comment 5: Lines 146-149: “the poloxamer used is not able to reticulate under 17 % w/v while sodium hyaluronate, chitosan and Carbopol® are able to jellify at much lower concentration” this must be supported by references. Why did you selected these polymers?

Answer to comment 5: Regarding the first part of the comment, the statement was changed as follow:

Pluronic® F127 is a poloxamer able form rigid gels at different concentrations and temperatures, being the temperature dependent of the concatenation of the polymer. In order for it to reticulate at 32°C (skin temperature), Pluronic® F-127 concentration should be at least of 15% w/v [29]. Contrary, this concentration is too high for Carbopol®, chitosan and sodium hyaluronate. For this reason, Pluronic® F-127 was kept constant throughout the experimental design

Regarding the second part of the comment, the polymer selection was mentioned in the discussion section (Lines 450-464).

Ref 29: Chaibundit, C.; Ricardo, N.M.P.S.; Costa, F. de M.L.L.; Yeates, S.G.; Booth, C. Micellization and Gelation of Mixed Copolymers P123 and F127 in Aqueous Solution. Langmuir 2007, 23, 9229–9236, doi:10.1021/la701157j.

Comment 6: Line 324. What is the value of Cur solubility in water?

Cur solubility was calculated to be 3×10-3 µg/mL by Farlconieri et al (Ref. 30). This information was included in the main text as follows:

As Cur is a polyphenol with extremely low solubility (3×10-3 µg/mL, [30]), a solubilizing strategy was necessary. In this case, cyclodextrins were employed.

Ref 30: Falconieri, M.C.; Adamo, M.; Monasterolo, C.; Bergonzi, M.C.; Coronnello, M.; Bilia, A.R. New Dendrimer-Based Nanoparticles Enhance Curcumin Solubility. Planta Med. 2017, 83, 420–425, doi:10.1055/s-0042-103161.

Comment 7: Line 413. Permeation studies. Why did you evaluate the permeation during 72 h? It is not reasonable to think that this formulation can stay on skin surface for a long time. Many studies document the non correspondence between the use of artificial and natural membranes. This experiment could not reproduce the real use conditions.

Answer to comment 7: Although most topical formulation are expected to be used without a support, it is not uncommon to have them in the form of patches or to occlude a gel/cream with a patch to increase residence time. For this reason, the experiment took place over 72 h.

As for the second comment of the Reviewer, the use of excised animal skin or tissues as natural membranes gives rise to problems of high variability and poor reproducibility of the results, owing to the great biological variability both inter- and intra-species. In addition, due to ethic reasons, the use of animal membranes should be as limited as possible. As a result, a growing interest in the development of artificial-membrane-based permeation systems has evolved (see for ex.:Di Cagno et al. (2015) and Berben et al. (2018)) 

Permeation studies performed with lipophilic artificial membranes simulating the biological ones showed a good predictive power of the fraction of drug absorbed in vivo, superior tan that provided by Caco-2 cells (Corti et al, (2006)). Moreover, artificial membranes impregnated with suitable lipids to simulate the skin barrier function allowed to obtain highly reproducible results, showing a similar ranking with respect to experiments performed with animal skin models (Mura et al., (2014) ; Engesland et al., (2013); Karadzovska and J.E. Riviere (2013)).

  • Di Cagno et al. New biomimetic barrier Permeapad™ for efficient investigation of passive permeability of drugs. Eur. J. Pharm. Sci. 73 (2015) 29-34.
  • Berben et al., Drug permeability profiling using cell-free permeation tools: Overview and
  • Eur. J. Pharm. Sci. 119 (2018) 219-233).
  • Corti, G.; Maestrelli, F.; Cirri, M.; Zerrouk, N.; Mura, P. Development and evaluation of an in vitro method for prediction of human drug absorption II. Demonstration of the method suitability. Eur. J. Pharm. Sci. 2006, 27, 354–362.
  • Mura et al., 2014 see ref. 76
  • Engesland, A., Skar, M., Hansen, T.,Škalko-Basnet, N., Flaten, G.E., New applications of phospholipid vesicle-based permeation assay: permeation model mimickingskin barrier. J. Pharm. Sci. 102 (2013) 1588–1600
  • Karadzovska and J.E. Riviere, Assessing vehicle effects on skin absorption using artificial membrane assays. Eur. J. Pharm. Sci. 50 (2013) 569-576

Comment 8: Line 553. modified dispersed amount method, a reference is necessary? Why the authors did not used official methods reported in Pharmacopoeia?

Answer to comment 8: The dispersed amount method is usually used in preformulation screening studies, when only small amounts of powders samples are available. Official methods reported in Pharmacopeia require high volumes of dissolution medium and then greater amounts of samples. A reference about the use of this method has been cited (ref 13)

Jug, M.; Kosalec, I.; Maestrelli, F.; Mura, P. Analysis of Triclosan Inclusion Complexes with β-Cyclodextrin and Its Water-Soluble Polymeric Derivative. J. Pharm. Biomed. Anal. 2011, 54, 1030–1039, doi:10.1016/j.jpba.2010.12.009

Comment 9: Par. 4.8.2. The release behavior of Cur from the hydrogel was evaluated by dialysis. This method is not acceptable to evaluate the release from a semisolid formulation. Official methods must be used for example Franz diffusion cell (USP).

Answer to comment 9: Franz diffusion cell is currently used for permeation (diffusion) studies and it is not an official method to evaluate drug release from a semisolid formulation.

No compendial standards exist at this regard, and drug release is currently assessed using a variety of methods, including dialysis membrane. A pertinent reference about the use of the dialysis method to evaluate the drug release from a hydrogel has been added (Hua et al., 2014, see ref 75).

Hua, S., Comparison of in vitro dialysis release methods of loperamide-encapsulated liposomal gel for topical drug delivery. Int. J. Nanomed. 2014:9 735–744

Comment 10: Line 597. “PBS pH 7.4”. Why did you used this pH? It is not applicable for formulations intended to be applied on skin.

Answer to comment 10: As Reviewer 3 pointed, pH 7.4 is not adequate for formulation applied on skin. However, the main objective of this method is not to reproduce skin conditions, but to study Cur behavior after trespassing stratum corneum. Because of the different pH, we suspected different behavior in both compartments, hence, the reason to perform two different assays at both pH.

Comment 11: Line 582. “After the binary system was as dissolved as possible” This phrase is not scientific

Answer to comment 11: the procedure for preparing the hydrogel was improved. The sentence changed as follow:

Cur/EpiβCD binary system was dissolved in 3.6 mL of Pluronic® 23.6 % w/v. After 30 mins, 1.4 mL of absolute ethanol was added, ensuring complete dissolution.

Comment 12: The English must be revised

Answer to comment 12: English has been revised.

Round 2

Reviewer 2 Report

The manuscript was revised as suggested.. The paper can be accepted as it is.